# DATA FUSION–ENHANCED DECISION TRANSFORMER FOR STABLE CROSS-DOMAIN GENERALIZATION

## ABSTRACT

Cross-domain shifts present a significant challenge for decision transformer (DT) policies. Existing cross-domain policy adaptation methods typically rely on a single simple filtering criterion to select source trajectory fragments and stitch them together. They match either state structure or action feasibility. However, the selected fragments still have poor stitchability: state structures can misalign, the return-to-go (RTG) becomes incomparable when the reward or horizon changes, and actions may jump at trajectory junctions. As a result, RTG tokens lose continuity, which compromises DT's inference ability. To tackle these challenges, we propose Data Fusion–Enhanced Decision Transformer (DFDT), a compact pipeline that restores stitchability. Particularly, DFDT fuses scarce target data with selectively trusted source fragments via a two-level data filter, maximum mean discrepancy (MMD) mismatch for state-structure alignment, and optimal transport (OT) deviation for action feasibility. It then trains on a feasibility-weighted fusion distribution. Furthermore, DFDT replaces RTG tokens with advantage-conditioned tokens, which improves the continuity of the semantics in the token sequence. It also applies a $Q$-guided regularizer to suppress junction value and action jumps. Theoretically, we provide bounds that tie state value and policy performance gaps to the MMD-mismatch and OT-deviation measures, and show that the bounds tighten as these two measures shrink. We show that DFDT improves return and stability over strong offline RL and sequence-model baselines across gravity, kinematic, and morphology shifts on D4RL-style control tasks, and further corroborate these gains with token-stitching and sequence-semantics stability analyses.

## 1 INTRODUCTION

Offline reinforcement learning (RL) aims to turn logged interaction data into deployable policies without further environment access, improving safety and sample efficiency in costly or risky domains (Levine et al., 2020). Sequence-modelling approaches such as the Decision Transformer (DT) recast RL as conditional sequence prediction and obtain strong performance by conditioning actions on return-to-go (RTG) (Chen et al., 2021; Janner et al., 2021a). However, when the training and test dynamics differ, DT-style policies often fail. At a high level, the underlying challenge is *dynamics shift* between source and target environments, whose token-level consequences include: stitched trajectory fragments losing continuity, state manifolds drifting across domains, actions that are feasible in the source dynamics becoming implausible at stitch junctions, and RTG tokens becoming incomparable under changes in the *distribution and scale* of returns and effective horizons (even when the reward function form is unchanged). As a result, DT agents can overfit source-domain statistics and generalize poorly under cross-domain deployment.

Cross-domain offline RL seeks to alleviate this issue by combining rich source logs with scarce target-domain data. Existing methods filter or reweight source transitions to bias learning toward samples that better match the target dynamics, using support-aware selection, representation-based filtering, stationary-distribution regularization, or optimal-transport (OT) alignment (Liu et al., 2024; Wen et al., 2024; Xue et al., 2023; Lyu et al., 2025b). These approaches are effective for value-based and actor–critic pipelines, but they operate at the level of occupancy or transition distributions rather than the sequence tokens governing DT policies. In particular, they do not explicitly control the

Figure 1: An overview of our proposed framework. Credible source fragments are first selected by an MMD-based state-structure gate and an OT-based action-feasibility reweighting, then fused with scarce target data and fed, together with advantage-conditioned tokens $A$, into a Decision Transformer whose attention heads predict stable actions $\hat{a}_t$ under cross-domain shifts.

*stitchability radii* that quantify state-structure and action-feasibility mismatches under dynamics shift, nor do they directly repair token semantics at trajectory junctions.

We propose Data Fusion–Enhanced Decision Transformer (DFDT), a compact cross-domain adaptation framework that explicitly restores token-level stitchability for DT policies. DFDT fuses target data with selectively trusted source fragments via a two-level filter: (i) an MMD-based state-structure gate that retains source fragments whose latent state trajectories stay close to the target manifold, and (ii) OT-derived scores that quantify action feasibility and define per-sample weights. These two components later correspond to state-structure and action-feasibility stitchability radii in our analysis. We train value and Q-functions on the resulting *feasibility-weighted fusion distribution*, and replace brittle RTG tokens with *advantage-conditioned tokens* that provide a reward- and horizon-agnostic conditioning signal. A lightweight Q-guided regularizer further suppresses value and action jumps at stitch junctions, yielding smoother token transitions.

Our contributions are threefold:

- We identify and characterize failure modes of DT under cross-domain stitching, and propose a two-level MMD+OT fragment filtering and weighting framework that defines a feasibility-weighted fusion distribution for cross-domain offline RL.

- We design DFDT, a DT-based policy adaptation method that combines feasibility-weighted critics, advantage-conditioned tokens, and a Q-guided regularizer to restore token-level stitchability under dynamics shifts.

- We provide theoretical bounds that decompose dynamics shift into state-structure and action-feasibility stitchability radii and tie value and performance gaps to these radii and estimation errors, and empirically show that DFDT attains strong performance and stable sequence semantics across gravity, kinematic, and morphology shifts.

Section 2 formalizes the cross-domain offline RL setting and recalls expectile regression, Section 3 presents our fusion framework and the DFDT algorithm, Section 5 reports experiments and diagnostics, and Section 6 concludes.

## 2 PRELIMINARIES AND PROBLEM FORMULATION

We begin by formalizing the cross-domain offline RL setting and introducing the notation used throughout the paper. Our focus is on adapting from a rich source dataset collected under dynamics $P_S$ to a target environment with different dynamics $P_T$, where only limited target data are available. This setting highlights how dynamics shifts break token-level trajectory stitching for standard Decision Transformers and motivates the design choices in DFDT.

**Cross-domain offline RL and token stitchability.** We consider two infinite-horizon Markov Decision Processes (MDPs), the source domain $\mathcal{M}_S := (\mathcal{S}, \mathcal{A}, P_S, r_S, \gamma, \rho_0)$ and the *target* domain $\mathcal{M}_T := (\mathcal{S}, \mathcal{A}, P_T, r_T, \gamma, \rho_0)$. The two domains share the same state space $\mathcal{S}$, action space $\mathcal{A}$,

reward function $r : \mathcal{S} \times \mathcal{A} \to \mathbb{R}$ (bounded by $r_{\max}$), discount factor $\gamma \in [0, 1)$, and initial distribution $\rho_0$, but differ in their transition kernels $P_S \neq P_T$. Let $\mathbb{P}_S$ and $\mathbb{P}_T$ denote the corresponding data occupancy distributions of datasets. For any MDP $\mathcal{M}$ and policy $\pi$, let the normalized discounted state and state-action occupancy measures be $d_{\mathcal{M}}^\pi(s) := (1 - \gamma) \sum_{t=0}^{\infty} \gamma^t P_{\mathcal{M}}^\pi(s \mid t)$ and $\nu_{\mathcal{M}}^\pi(s, a) := d_{\mathcal{M}}^\pi(s) \, \pi(a \mid s)$, and define the performance of $\pi$ by $J_{\mathcal{M}}(\pi) := \mathbb{E}_{(s,a) \sim \nu_{\mathcal{M}}^\pi}[r(s, a)]$. Let $\mathcal{D}_{\mathrm{src}} = \{(s, a, r, s')\}$ be an offline dataset from $\mathcal{M}_S$ and $\mathcal{D}_{\mathrm{tar}} = \{(s, a, r, s')\}$ a much smaller dataset from $\mathcal{M}_T$. We aim to learn a policy $\pi^\star$ that maximises $J_T(\pi)$ without online interaction with $\mathcal{M}_T$. The core challenge is the cross-domain shift $P_S \neq P_T$, which is especially harmful to DT policies that rely on token-level continuity in RTG, state, and action: source and target state manifolds misalign, RTG becomes incomparable under reward and horizon shifts, and actions feasible under $P_S$ can be implausible under $P_T$, creating stitch discontinuities. This breaks local sequence structure, induces exposure bias, and destabilizes RTG conditioning; using only $\mathcal{D}_{\mathrm{src}}$ extrapolates invalid next tokens under $P_T$, while $\mathcal{D}_{\mathrm{tar}}$ alone lacks coverage. We therefore use $\mathcal{D}_{\mathrm{src}}$ and $\mathcal{D}_{\mathrm{tar}}$ to restore token continuity, aligning state structure, ensuring action feasibility, and replacing fragile RTG with a stable conditioning signal, hence Transformer policies generalize across domains while controlling distributional mismatch.

**Expectile regression and advantage estimation.** For a response $Y \in \mathbb{R}$ and covariates $X \in \mathcal{X}$, the $\zeta$-expectile regression function ($\zeta \in (0, 1)$) is the map $m_\zeta : \mathcal{X} \to \mathbb{R}$ that minimizes the objective $\mathbb{E}\big[\rho_\zeta\big(Y - m_\zeta(X)\big)\big]$ with $\rho_\zeta(u) = \big|\zeta - \mathbf{1}\{u < 0\}\big| u^2$, yielding a unique minimizer under mild integrability by strict convexity. Expectiles continuously interpolate tail emphasis: $\zeta = \frac{1}{2}$ recovers the conditional mean, while $\zeta \to 1$ (resp. $\to 0$) increases sensitivity to upper (resp. lower) tails.

## 3 DFDT: Data Fusion–Enhanced Decision Transformer

We now present DFDT, a framework that restores token-level stitchability under dynamic shifts by combining data fusion, advantage-conditioned tokenization, and Q-regularized DT training. At a high level, DFDT constructs a feasibility-weighted fusion distribution over source and target fragments, uses shared advantages instead of RTG as conditioning tokens, and trains a DT policy that is regularized to avoid large value and action jumps at fragment junctions. This section proceeds from data to model: Section 3.1 describes the fusion distribution, Section 3.2 introduces advantage-conditioned tokenization, and Section 3.3 specifies the training objective.

### 3.1 Two-level fragment filtering and feasibility-weighted fusion

Before sequence modeling, DFDT applies two-level filtering and reweighting to source fragments. At the state-structure level, we use an MMD criterion in a learned latent space to remove fragments whose transitions are incompatible with the target manifold. At the action-feasibility level, we use an OT-based cost over state–action pairs to assign feasibility weights, down-weighting fragments whose actions are implausible under target dynamics and forming a feasibility-weighted fusion distribution that blends source and target data.

#### 3.1.1 MMD-based fragment selection

To enforce state-structure compatibility, DFDT compares source and target fragments in a latent representation space using an MMD criterion. Fragments whose latent state transitions lie far from those observed in the target data are discarded or strongly down-weighted, yielding a subset of source trajectories whose state evolution better aligns with target dynamics and reduces abrupt state jumps at stitch junctions.

Let $z = f_\phi(s)$ be a shared encoder that extracts state features. For a source fragment $\tau^S = (s_1^S, a_1^S, r_1^S, \ldots, s_n^S, a_n^S, r_n^S)$ and a target fragment $\tau^T = (s_1^T, a_1^T, r_1^T, \ldots, s_m^T, a_m^T, r_m^T)$, we compute the RBF-kernel MMD in latent space:

$$\mathrm{MMD}_k^2(\tau^S, \tau^T) = \frac{1}{n^2} \sum_{i,j} k(z_i^S, z_j^S) + \frac{1}{m^2} \sum_{i,j} k(z_i^T, z_j^T) - \frac{2}{nm} \sum_{i,j} k(z_i^S, z_j^T), \quad (1)$$

where $z_i^S = f_\phi(s_i^S)$ and $k$ is an RBF kernel. This score measures structural similarity between dynamics; we keep the top-$\xi\%$ source fragments with the smallest values to form a pseudo-target

buffer $\mathcal{D}_{\text{sim}}$. We then define the state-structure MMD distance of $\tau^S$ to the target dataset:

$$d^m(\tau^S) \ = \ \mathbb{E}_{\tau^T} \text{MMD}_k\big(\{z_i^S = f_\phi(s_i^S)\}_{i=1}^\ell, \ \{z_j^T = f_\phi(s_j^T)\}_j^\ell\big), \tag{2}$$

and the hard gate $I_m(\tau^S) := \mathbf{1}\big(d^m(\tau^S) \le q_\xi\big)$ that retains the top-$\xi\%$ most similar fragments.

### 3.1.2 OT-BASED ACTION FEASIBILITY AND WEIGHTS

While MMD ensures that selected source fragments lie on a target-like state manifold, it does not guarantee that their actions remain feasible under target dynamics. We therefore introduce an OT-based notion of action feasibility over state–action pairs and convert the resulting costs into feasibility weights that modulate each fragment's influence in both critic and DT training. We note that the OT-based feasibility score directly reuse the discrete OT formulation proposed in OTDF (Lyu et al., 2025a). In DFDT, we treat this as a well-validated building block and focus our contributions on how these OT weights are embedded into the two-level fusion framework and sequence-model training.

For each $\tau^S \in \mathcal{D}_{\text{src}}$, we use optimal transport (Villani et al., 2008; Peyré et al., 2019) to evaluate how plausible its actions are under target dynamics. Define concatenations $v_t^S = s_t^S \oplus a_t^S \oplus r_t^S \oplus s_{t+1}^S$ and $v_t^T = s_t^T \oplus a_t^T \oplus r_t^T \oplus s_{t+1}^T$, with $v_S^S \sim \mathcal{D}_{\text{src}}$ and $v_t^T \sim \mathcal{D}_{\text{tar}}$. Given a 1-Lipschitz cost function $C$ and coupling $\mu$, the Wasserstein distance is defined as

$$W_1 = \min_{\mu \in \mathcal{M}} \sum_{t=1}^{|\mathcal{D}_{\text{src}}|} \sum_{t'=1}^{|\mathcal{D}_{\text{tar}}|} C(v_t^S, v_{t'}^T) \, \mu_{t,t'}. \tag{3}$$

Suppose solving the optimization problem in Eq. 3 gives the OT coupling $\mu^*$ (Kantorovich, 1942); we determine the deviation between a source-domain sample and the target dataset via

$$d^w(u_t) = -\sum_{t'=1}^{|\mathcal{D}_{\text{tar}}|} C(v_t^S, v_{t'}^T) \, \mu_{t,t'}^*, \tag{4}$$

which becomes larger when the source sample aligns well with the target behavior (i.e., lower transport cost), and smaller otherwise. This OT credibility score prioritizes source actions whose transport cost to the target behavior is small, improving action continuity across stitched tokens. In other words, MMD bounds how far the selected source state transitions can deviate from the target dynamics manifold, whereas OT bounds how far the weighted source state–action pairs can deviate from target-feasible behavior.

### 3.1.3 FEASIBILITY-WEIGHTED FUSION DISTRIBUTION

Combining the state-level MMD filter and the action-level OT weights yields a feasibility-weighted source distribution $\mathbb{P}_S^w$ over the origin one $\mathbb{P}_S$. We then define a fusion distribution $\mathbb{P}_{\text{mix}}^w$ that mixes this weighted source distribution with the target distribution according to a tunable mixing ratio:

**Definition 3.1** (Two-level feasibility-weighted data fusion framework). For each triple $u = (s, a, s')$ from a gated source fragment, let $d^w(u)$ be the OT action feasibility score, and set the raw per-sample weight

$$w(u) := I_m(\tau^S) \exp\big(\eta_w \, d^w(u)\big),$$

where $\eta_w > 0$ is the weight temperature coefficient. Normalize the raw per-sample weight as $\tilde{w}(u) := \frac{w(u)}{\mathbb{E}_{u \sim \mathbb{P}_S}[w(u)]}$. Define the weighted source and target–source data fusion distributions as

$$\mathbb{P}_S^w(u) \propto \tilde{w}(u) \, \mathbb{P}_S(u) \quad \text{and} \quad \mathbb{P}_{\text{mix}}^w = (1 - \beta) \, \mathbb{P}_T + \beta \, \mathbb{P}_S^w, \quad \beta \in [0, 1].$$

This fusion distribution is the backbone of DFDT: critics are trained under $\mathbb{P}_{\text{mix}}^w$ and DT policies also receive trajectories sampled from $\mathbb{P}_{\text{mix}}^w$, keeping value learning and sequence modeling consistent.

**Learning objective under feasibility-weighted data fusion distribution.** We propose that training samples are drawn from $\mathbb{P}_{\text{mix}}^w$ and we minimize weighted TD losses for $Q$ and $V$, respectively

$$\mathcal{L}_V = \mathbb{E}_{\mathbb{P}_{\text{mix}}^w}\Big[\big(r(s,a) + \gamma V(s') - V(s)\big)^2\Big], \quad \mathcal{L}_Q = \mathbb{E}_{\mathbb{P}_{\text{mix}}^w}\Big[\big(r + \gamma V(s') - Q(s,a)\big)^2\Big].$$

We train the policy with a weighted DT objective $\mathcal{L}_{\text{DT}}^w$ under $\mathbb{P}_{\text{mix}}^w$ and a $Q$-regularized term:

$$\mathcal{L}_\pi = \mathcal{L}_{\text{DT}}^w - \alpha \, \mathbb{E}_{\mathbb{P}_{\text{mix}}^w}\big[Q(s, \pi(s))\big].$$

## 3.2 Advantage-Conditioned Tokenization for Trajectory Stitching

Even with a stitchable fusion distribution, standard DT suffers from brittle RTG tokens: returns are sensitive to reward scaling and can exhibit sharp discontinuities when stitching fragments across domains. DFDT instead uses advantage-conditioned tokens derived from shared value and $Q$-functions trained under $\mathbb{P}_{\text{mix}}^w$, which yields more stable supervision at fragment junctions.

**Shared value and $Q$-functions.** We train a shared value function $V_\varphi$ and $Q$-function $Q_\psi$ so that advantages are comparable across domains and can guide tokenization. Since selected fragments are structurally aligned with the target dynamics, the shared $V_\varphi$ promotes a consistent reward structure across domains. Specifically, we estimate $V_\varphi$ via a weighted expectile regression:

$$\mathcal{L}_\varphi = \mathbb{E}_{\tau^T} \left[ \rho_\zeta(\Delta_{V_\varphi}) \right] + \mathbb{E}_{\tau^S} \left[ \exp(\eta_w \, d^w) \, I_m(\tau^S) \, \rho_\zeta(\Delta_{V_\varphi}) \right], \tag{5}$$

and

$$\mathcal{L}_\psi = \mathbb{E}_{\tau^T} \left[ \rho_{\frac{1}{2}}(\Delta_{Q_\psi}) \right] + \mathbb{E}_{\tau^S} \left[ \exp(\eta_w \, d^w) \, I_m(\tau^S) \, \rho_{\frac{1}{2}}(\Delta_{Q_\psi}) \right], \tag{6}$$

where $\rho_{\frac{1}{2}} := \rho_{\zeta=\frac{1}{2}}$ and $I_m(\tau^S) := \mathbf{1}\big(d^m(\tau^S) \le q_\xi\big)$ is the hard gate from Definition 3.1. The expectile function $\rho_\zeta$ is defined in Section 2. The one-step TD residuals are defined as $\Delta_V(s, a, s') := r(s, a) + \gamma V(s') - V(s)$ and $\Delta_Q(s, a, s') := r(s, a) + \gamma V(s') - Q(s, a)$.

**Sequence format with advantage tokens.** For each state–action pair $(s_i, a_i)$ in a selected fragment $\tau = (s_t, a_t, r_t, \ldots, s_{t+k}, a_{t+k}, r_{t+k})$, we generate a pseudo-return token as $A(s_i, a_i) = Q_\psi(s_i, a_i) - V_\varphi(s_i)$. These weighted advantage values replace the original (unavailable or inconsistent) RTG signals. The resulting transformer token sequence

$$(s_t, a_t, A_t, \ldots, s_{t+k}, a_{t+k}, A_{t+k}) \tag{7}$$

forms the input for policy training, enforcing reward continuity and structure-aware return alignment and enabling stable cross-domain adaptation. We introduce a command network $C_\omega$ that produces an *advantage-consistent command token* that serves as an RTG replacement during inference. This network is trained using supervised learning with the advantage information $A$ of the dataset.

## 3.3 Weighted Q-Regularized Transformer Training

We finally specify how DFDT trains its critics and DT policy under the feasibility-weighted fusion distribution $\mathbb{P}_{\text{mix}}^w$. At a high level, we combine (i) critic updates based on multi-step TD targets sampled from $\mathbb{P}_{\text{mix}}^w$, (ii) a weighted sequence-modeling loss on advantage-conditioned trajectories, and (iii) a Q-regularizer that explicitly suppresses value and action jumps at fragment junctions. Algs. 1 and 2 summarize the full training and inference procedures.

**Critic network update.** The parameters $\phi_i$ ($i \in \{1, 2\}$) of the twin critics $Q_{\phi_i}$ are trained by minimizing

$$\mathcal{L}_\phi = \mathbb{E}_{\mathbb{P}_{\text{mix}}^w} \left[ \sum_{i=t-K+1}^{t-1} \big(\hat{Q}_i - Q_{\phi_i}(s_i, a_i)\big)^2 \right], \tag{8}$$

where $\hat{Q}_i = \sum_{j=i}^{t-1} \gamma^{j-i} r_j + \gamma^{t-i} \min_{k=1,2} Q_{\phi'_k}(s_t, \hat{a}_t)$ is a multi-step TD target and $Q_{\phi'_k}, \pi_{\theta'}$ are the target critic and policy. Writing out the definition of $\mathbb{P}_{\text{mix}}^w$ (Def. 3.1), Eq. (8) coincides with $\mathcal{L}_\phi(\phi; \mathcal{D}_{\text{tar}}) + \mathbb{E}_{\tau \sim \mathcal{D}_{\text{src}}, \hat{a}_t \sim \pi_{\theta'}} \big[ I_m(\tau^S) \sum_{i=t-K+1}^{t-1} \exp(\eta_w d_t^w) (\hat{Q}_i - Q_{\phi_i}(s_i, a_i))^2 \big]$, i.e., the standard Bellman loss on $\mathcal{D}_{\text{tar}}$ plus a feasibility-weighted contribution from source trajectories.

**Cross-domain DT loss.** The conditional DT policy receives the relabeled sequences and predicts the next action at each step. Its behavior cloning objective under the fusion distribution is

$$\mathcal{L}_{\text{DT}}^w(\theta) = \mathbb{E}_{\mathbb{P}_{\text{mix}}^w} \left[ \frac{1}{K} \sum_{i=t-K+1}^{t} \big(a_i - \pi_\theta(\tau)_i\big)^2 \right]. \tag{9}$$

Equivalently, expanding $\mathbb{P}_{\text{mix}}^w$ over $\mathcal{D}_{\text{tar}} \cup \mathcal{D}_{\text{src}}$ shows that $\mathcal{L}_{\text{DT}}^w(\theta) = \mathcal{L}_{\text{DT}}(\theta; \mathcal{D}_{\text{tar}}) + \mathbb{E}_{\tau^S \sim \mathcal{D}_{\text{src}}} \big[ K^{-1} I_m(\tau^S) \sum_{i=t-K+1}^{t} \exp(\eta_w d_i^w) (a_i - \pi_\theta(\tau^S)_i)^2 \big]$, where $\mathcal{L}_{\text{DT}}(\theta; \mathcal{D}_{\text{tar}})$ is the standard DT loss on the target data.

**Integrating $Q$-regularizer into the DT loss.** To bias the policy toward high-value actions and improve action-level stitching, we augment the DT objective with a $Q$-value regularizer:

$$\mathcal{L}_\pi = \mathcal{L}_{\text{DT}}^w - \alpha\, \mathbb{E}_{\mathbb{P}_{\text{mix}}^w}\left[\frac{1}{K}\sum_{i=t-K+1}^{t} Q_\phi(s_i, \pi(s_i))\right] - \eta_{\text{reg}}\,\mathcal{L}_{\text{reg}}(\pi), \tag{10}$$

where $Q_\phi(s,a) := \min_{i=1,2} Q_{\phi_i}(s,a)$ and $\mathcal{L}_{\text{reg}}(\pi)$ is a KL penalty between $\pi$ and the behavior policy $\pi_{\text{tar}}$ induced by $\mathcal{D}_{\text{tar}}$, with weight $\eta_{\text{reg}} > 0$. Using again the explicit form of $\mathbb{P}_{\text{mix}}^w$, the $Q$-regularizer term $-\alpha\, \mathbb{E}_{\mathbb{P}_{\text{mix}}^w}[K^{-1}\sum_i Q_\phi(s_i, \pi(s_i))]$ can be written as $-\alpha\, \mathbb{E}_{\tau \in \mathcal{D}_{\text{tar}} \cup \mathcal{D}_{\text{src}}}[K^{-1} I_m(\tau^S)\sum_{i=t-K+1}^{t}\exp(\eta_w d_i^w)\, Q_\phi(s_i, \pi(s_i))]$, highlighting that the same feasibility weights shape the regularization on both target and (filtered) source data.

## 4 THEORETICAL ANALYSIS

We analyze DFDT from a theoretical perspective by relating its feasibility-weighted fusion distribution and learned critics to target-domain performance. We introduce stitchability radii that quantify mismatches between the fusion and target distributions and show how these radii, together with critic approximation errors, control the resulting value and performance gaps.

### 4.1 SETUP AND ASSUMPTIONS

We formalize the feasibility-weighted source distribution $\mathbb{P}_S^w$, the fusion distribution $\mathbb{P}_{\text{mix}}^w$, and the target Bellman operator, and specify the function classes used to approximate value and $Q$-functions. Under standard boundedness and approximation assumptions, we characterize the errors incurred when learning critics under $\mathbb{P}_{\text{mix}}^w$ instead of the target distribution. These definitions and assumptions underpin the value and performance bounds below.

We define the estimation errors $\varepsilon_V, \varepsilon_Q$ as the conditional residual bounds under $\mathbb{P}_{\text{mix}}^w$:

$$\varepsilon_V := \sup_{s \in \mathcal{S}}\left|\mathbb{E}_{\mathbb{P}_{\text{mix}}^w}\big[\Delta_V(s,a,s')\,\big|\,s\big]\right|, \qquad \varepsilon_Q := \sup_{(s,a) \in \mathcal{S} \times \mathcal{A}}\left|\mathbb{E}_{\mathbb{P}_{\text{mix}}^w}\big[\Delta_Q(s,a,s')\,\big|\,s,a\big]\right|.$$

**Assumption 4.1** (Finite estimation errors). Due to finite samples and function class complexity, the fitted estimators satisfy finite errors, i.e., $\varepsilon_V, \varepsilon_Q < \infty$.

**Assumption 4.2** (Approximate fiber-constancy of $V$). Let $f_\phi$ be an encoder that maps inputs $s$ (either full states in a fully observed MDP, or observation histories in a partially observed setting) from $\mathcal{S}$ to latent codes in $\mathcal{Z}$. For any $z \in \mathcal{Z}$, the fiber (preimage) of $f_\phi$ over $z$ is the level set $f_\phi^{-1}(\{z\}) = \{s \in \mathcal{S} : f_\phi(s) = z\}$. The value varies little within the fibers of $f_\phi$: there exists $\varepsilon_H \geq 0$ such that

$$\sup\left\{|V(s) - V(\tilde{s})| : s, \tilde{s} \in \mathcal{S},\ f_\phi(s) = f_\phi(\tilde{s})\right\} \leq \varepsilon_H.$$

**Assumption 4.3** (Occupancy concentrability). Let $d_T^*$ and $d_T^{\text{mix}}$ denote the normalized discounted state-occupancy measures of $\pi_T^*$ and $\pi_{\text{mix}}$ in the target MDP. Assume absolute continuity and a finite essential bound on the density ratio: $d_T^* \ll d_T^{\text{mix}}$ and $\kappa := \|\frac{\text{d}d_T^*}{\text{d}d_T^{\text{mix}}}\|_\infty < \infty$.

### 4.2 STITCHABILITY RADII AND PERFORMANCE GUARANTEE

To quantify how well the feasibility-weighted fusion distribution $\mathbb{P}_{\text{mix}}^w$ can mimic target behavior, we introduce stitchability radii that bound an MMD-based state-structure discrepancy and an OT-based state–action discrepancy between $\mathbb{P}_{\text{mix}}^w$ and the target distribution. Small radii mean that, after two-level filtering and weighting, fused fragments behave similarly to target trajectories in both state transitions and action feasibility. Using these quantities, we derive a performance bound in which the return gap between DFDT and an ideal target policy is controlled by critic errors, the two radii, and standard finite-sample terms; relying only on state-level or only on action-level matching would leave one radius uncontrolled and weaken this guarantee.

**Definition 4.1** (Stitchability radii). Let $\Delta_m := \sup_{\tau^S : I_m(\tau^S)=1} d^m(\tau^S)$ and $\Delta_w := W_1(\mathbb{P}_S^w, \mathbb{P}_T)$, which measure the state-structure MMD and 1-Wasserstein distances between the weighted source triples and the target domain, respectively.

Intuitively, one may view $\mathbb{P}_S^w$ and $\mathbb{P}_T$ as being induced by (possibly unknown) behavior policies in the source and target domains. Then, our performance bound is expressed directly in terms of the resulting stitchability radii and concentrability coefficient of occupancy distribution.

**Theorem 4.1** (Performance bound under stitchability radii). *Under Assumptions 4.1–4.3, training with $\mathbb{P}_{\mathrm{mix}}^w$ yields estimators $V$ and $Q$. Let $V_T$ and $Q_T$ be the state and state–action value functions learned from the target dataset. Let $\pi_T^*$ and $\pi_{\mathrm{mix}}$ denote any optimal policies learned from the target MDP $\mathbb{P}_T$ and the mixed MDP $\mathbb{P}_{\mathrm{mix}}^w$, respectively. Let $d_T^* \otimes \pi_T^*$ be the normalized discounted state–action occupancy of $\pi_T^*$ under $\mathbb{P}_T$ and $\Delta_\pi := \mathbb{E}_{s \sim \mathbb{P}_T}\left[\|\pi_T^*(\cdot|s) - \pi_{\mathrm{mix}}(\cdot|s)\|_1\right]$. Then, for some constants $C_1, C_2, C_3, C_H, C_\pi > 0$,*

$$\|V - V_T\|_{1,\mathbb{P}_T} \leq \frac{C_1\,\beta\,(\Delta_m + \Delta_w) + 2\beta\,\varepsilon_H + \varepsilon_V}{1 - \gamma}, \tag{11}$$

$$\|Q - Q_T\|_{1,\mathbb{P}_T} \leq \frac{C_2\,\beta\,(\Delta_m + \Delta_w) + 2\beta\,\varepsilon_H + \varepsilon_Q}{1 - \gamma}. \tag{12}$$

*Moreover, by a performance-difference bound,*

$$J_T(\pi_T^*) - J_T(\pi_{\mathrm{mix}}) \leq \frac{C_3(1+\kappa)}{(1-\gamma)^2}\left(\beta(\Delta_m + \Delta_w) + C_H\,\beta\,\varepsilon_H + \varepsilon_V\right) + \frac{C_\pi}{(1-\gamma)^2}\,\Delta_\pi. \tag{13}$$

The proof is provided in Appendix E. In words, Theorem 4.1 is best read as a *decomposition* rather than a tight numerical bound: the performance gap splits into a standard residual policy-approximation term $\Delta_\pi$ (not controlled by our MMD/OT machinery) and a data-fusion term that depends on the stitchability radii $(\Delta_m, \Delta_w)$ and critic errors $(\varepsilon_V, \varepsilon_Q)$ induced by $\mathbb{P}_{\mathrm{mix}}^w$. If (i) our critics are not too misfitted ($\varepsilon_V, \varepsilon_Q$ small) and (ii) the MMD and OT distances $(\Delta_m, \Delta_w)$ between weighted source and target triples are small, then the value and performance gaps are controlled; using only MMD or only OT leaves the other radius uncontrolled, explaining why we need both.

# 5 EXPERIMENTS

In this section, we evaluate DFDT under gravity, kinematic, and morphology shifts, centring on two research questions (RQ): (1) Does DFDT outperform strong prior baselines across gravity, kinematic, and morphology shifts and across source and target dataset qualities? (2) Can DFDT provide stable sequence semantics for policy adaptation? We also refer the reader to Appendix I for a detailed ablation study.

## 5.1 EXPERIMENTAL SETUP

**Tasks and datasets.** We evaluate policy adaptation under three dynamics shifts, *gravity*, *kinematics*, and *morphology*, on four MuJoCo tasks (HalfCheetah, Hopper, Walker2d, Ant) in OpenAI Gym Brockman et al. (2016). Gravity scales the magnitude of $g$; kinematics constrains joint ranges; morphology changes link dimensions. We adopt the configurations of Lyu et al. (2025b). The setting is cross-domain offline RL: abundant source data but scarce target data from shifted environments. Sources are D4RL "-v2" datasets (medium, medium-replay, medium-expert) (Fu et al., 2020); targets are the D4RL-style datasets of Lyu et al. (2025b) (medium / medium-expert / expert), each with 5,000 transitions. This low-data regime, known to challenge standard offline RL Liu et al. (2024); Wen et al. (2024); Lyu et al. (2025b), yields 108 tasks across the three shift families.

**Baselines.** We compare DFDT with strong offline RL: IQL (Kostrikov et al., 2022) (expectile value regression with advantage-weighted policy), and sequence-modelling baselines for cross-domain adaptation: DT (Chen et al., 2021) (return-to-go sequence model), QT (Hu et al., 2024) (value-aware DT), and a DADT variant (Kim et al., 2022) with dynamics-aware tokenisation but no filtering. We also include recent cross-domain methods: DARA (Liu et al., 2022), IGDF (Wen et al., 2024), and OTDF (Lyu et al., 2025b), covering reward reweighting, representation filtering, and OT-based data fusion. For all baselines, we use the authors' recommended hyperparameters and code, modifying only the dataset and environment identifiers.

**Evaluation protocol.** We adopt the cross-domain setup in Sec. 2, using abundant D4RL source logs (*medium*, *medium-replay*, *medium-expert*) and scarce target logs (*medium*, *medium-expert*, *expert*)

Table 1: **Performance comparison of cross-domain offline RL algorithms under morphology shifts.** half = halfcheetah, hopp = hopper, walk = walker2d, m = medium, r = replay, e = expert. The 'Target' column indicates target-domain offline data quality. We report *normalized* target-domain performance (*mean ± std.*) across source qualities {*medium, medium-replay, medium-expert*} and target qualities {*medium, medium-expert, expert*}, averaged over **five** seeds; best per row is highlighted.

| Source | Target | IQL | DARA | IGDF | OTDF | DT | QT | DADT | DFDT |
|---|---|---|---|---|---|---|---|---|---|
| half-m | medium | 30.0 | 26.6 | 41.6 | 39.1 | 34.6 | 34.5 | 34.8 | **44.2**±0.1 |
| half-m | medium-expert | 31.8 | 32.0 | 29.6 | 35.6 | 30.8 | −1.3 | 36.5 | **42.5**±1.9 |
| half-m | expert | 8.5 | 9.3 | 10.0 | 10.7 | 4.7 | 0.8 | 11.5 | **69.0**±7.3 |
| half-m-r | medium | 30.8 | 35.6 | 28.0 | 40.0 | 30.3 | 31.1 | 30.2 | **42.9**±2.0 |
| half-m-r | medium-expert | 12.9 | 16.9 | 12.0 | 34.4 | 19.4 | 24.6 | 25.7 | **42.8**±0.6 |
| half-m-r | expert | 5.9 | 3.7 | 5.3 | 8.2 | 4.7 | 11.3 | 9.5 | **53.0**±18.7 |
| half-m-e | medium | 41.5 | 40.3 | 40.9 | 41.4 | 34.9 | 22.2 | 36.4 | **44.4**±0.1 |
| half-m-e | medium-expert | 25.8 | 30.6 | 26.2 | 35.1 | 36.5 | 20.7 | 37.1 | **43.8**±0.5 |
| half-m-e | expert | 7.8 | 8.3 | 7.5 | 9.8 | 7.7 | 7.6 | 5.4 | **73.7**±7.0 |
| hopp-m | medium | 13.5 | 13.5 | 13.4 | 11.0 | 12.1 | 10.1 | 11.4 | **44.7**±16.5 |
| hopp-m | medium-expert | 13.4 | 13.6 | 13.3 | 12.6 | 13.2 | 13.2 | 13.1 | **36.0**±20.5 |
| hopp-m | expert | 13.5 | 13.6 | 13.9 | 10.7 | 12.9 | 13.1 | 13.5 | **56.1**±39.1 |
| hopp-m-r | medium | 10.8 | 10.2 | 12.0 | 8.7 | 13.3 | 13.1 | 14.4 | **53.2**±21.5 |
| hopp-m-r | medium-expert | 11.6 | 10.4 | 8.2 | 9.7 | 12.4 | 15.6 | 12.2 | **79.9**±13.0 |
| hopp-m-r | expert | 9.8 | 9.0 | 11.4 | 10.7 | 12.7 | **15.7** | 13.7 | **15.7**±2.7 |
| hopp-m-e | medium | 12.6 | 13.0 | 12.7 | 7.9 | 11.8 | 9.9 | 11.9 | **93.5**±4.7 |
| hopp-m-e | medium-expert | 14.1 | 13.8 | 13.3 | 9.6 | 11.8 | 12.6 | 10.7 | **69.7**±27.3 |
| hopp-m-e | expert | 13.8 | 12.3 | 12.8 | 5.9 | 12.0 | 12.7 | 11.7 | **86.5**±21.4 |
| walk-m | medium | 23.0 | 23.3 | 27.5 | **50.5** | 23.7 | 11.5 | 20.8 | 46.6±9.3 |
| walk-m | medium-expert | 21.5 | 22.2 | 20.7 | 44.3 | 22.4 | 29.0 | 25.3 | **41.1**±5.3 |
| walk-m | expert | 20.3 | 17.3 | 15.8 | 55.3 | 15.6 | 23.8 | 28.3 | **70.3**±22.1 |
| walk-m-r | medium | 11.3 | 10.9 | 13.4 | 37.4 | 12.3 | 30.1 | 28.3 | **44.8**±5.0 |
| walk-m-r | medium-expert | 7.0 | 4.5 | 6.9 | 33.8 | 6.0 | 1.6 | 13.6 | **40.6**±20.7 |
| walk-m-r | expert | 6.3 | 4.5 | 5.5 | 41.5 | 10.1 | 1.1 | 9.5 | **86.3**±16.1 |
| walk-m-e | medium | 24.1 | 31.7 | 27.5 | 49.9 | 17.8 | 19.7 | 27.7 | **51.4**±9.2 |
| walk-m-e | medium-expert | 27.0 | 23.3 | 25.3 | **40.5** | 14.3 | 24.2 | 25.2 | 28.4±6.3 |
| walk-m-e | expert | 22.4 | 25.2 | 24.7 | 45.7 | 10.2 | 21.8 | 26.7 | **85.5**±9.9 |
| ant-m | medium | 38.7 | 41.3 | 40.9 | 39.4 | 37.9 | 38.6 | 42.5 | **42.6**±0.6 |
| ant-m | medium-expert | 47.0 | 43.3 | 44.4 | 58.3 | 48.1 | 1.0 | 44.0 | **75.4**±6.2 |
| ant-m | expert | 36.2 | 48.5 | 41.4 | 85.4 | 22.8 | −1.0 | 23.7 | **85.5**±11.8 |
| ant-m-r | medium | 38.2 | 38.9 | 39.7 | 41.2 | 17.5 | 25.0 | 37.8 | **41.4**±1.3 |
| ant-m-r | medium-expert | 38.1 | 33.4 | 37.3 | 50.8 | 28.6 | 8.2 | 39.0 | **78.3**±8.9 |
| ant-m-r | expert | 24.1 | 24.5 | 23.6 | 67.2 | 21.2 | 8.3 | 25.9 | **75.0**±15.8 |
| ant-m-e | medium | 32.9 | 40.2 | 36.1 | 39.9 | 41.3 | 35.1 | 27.4 | **42.0**±0.5 |
| ant-m-e | medium-expert | 35.7 | 36.5 | 30.7 | 65.7 | 57.3 | 12.8 | 43.1 | **69.5**±11.4 |
| ant-m-e | expert | 36.1 | 34.6 | 35.2 | **86.4** | 37.9 | 12.3 | 31.1 | 81.9±6.9 |
| | Total Score | 798.0 | 816.8 | 808.7 | 1274.3 | 760.8 | 570.6 | 859.6 | **2078.2** |

collected under gravity, kinematic, and morphology shifts. We report normalized target-domain returns (*mean ± std.*) over *five* seeds while sweeping all {*source quality*} × {*target quality*} pairs (3×3) across *halfcheetah*, *hopper*, *walker2d*, and *ant*. The computing method of normalized returns is described in Sec. F. All methods train offline on the prescribed source and target logs. More detailed experimental settings can be found in Sec. F.

## 5.2 MAIN RESULTS: RETURNS UNDER DYNAMICS SHIFTS (ANSWER RQ(1))

We train our method for 100k gradient updates with five random seeds and report normalized target-domain scores. Summary comparisons of DFDT against baselines under morphology and kinematic shifts are given in Tables 1 and 2, respectively; results for the gravity shifts are deferred to Appendix H due to space limit.

Table 2: **Performance comparison of cross-domain offline RL algorithms under kinematic shifts.** Abbreviations are as in Table 1. We report normalized target-domain performance (*mean ± std.*) over **five** seeds; best per row is highlighted.

| Source | Target | IQL | DARA | IGDF | OTDF | DT | QT | DADT | DFDT |
|--------|--------|-----|------|------|------|----|----|------|------|
| half-m | medium | 12.3 | 10.6 | 23.6 | 40.2 | 32.1 | 14.6 | 14.5 | **41.2**±0.5 |
| half-m | medium-expert | 10.8 | 12.9 | 9.8 | 10.1 | 22.4 | 6.2 | 21.4 | **40.8**±1.5 |
| half-m | expert | 12.6 | 12.1 | 12.8 | 8.7 | 13.9 | 5.0 | 15.8 | **27.5**±5.0 |
| half-m-r | medium | 10.0 | 11.5 | 11.6 | 37.8 | 11.6 | 10.7 | 8.8 | **40.8**±0.3 |
| half-m-r | medium-expert | 6.5 | 9.2 | 8.6 | 9.7 | 7.5 | 40.1 | 6.0 | **41.4**±1.6 |
| half-m-r | expert | 13.6 | 14.8 | 13.9 | 7.2 | 2.7 | 19.2 | 5.7 | **27.6**±7.4 |
| half-m-e | medium | 21.8 | 25.9 | 21.9 | 30.7 | 17.5 | 18.7 | 14.5 | **41.2**±0.9 |
| half-m-e | medium-expert | 7.6 | 9.5 | 8.9 | 10.9 | 13.1 | 3.7 | 11.4 | **35.5**±12.1 |
| half-m-e | expert | 9.1 | 10.4 | 10.7 | 3.2 | 19.5 | 10.3 | 19.4 | **26.0**±14.2 |
| hopp-m | medium | 58.7 | 43.9 | 65.3 | 65.6 | 16.4 | 19.7 | 3.6 | **66.5**±0.9 |
| hopp-m | medium-expert | **68.5** | 55.4 | 51.1 | 55.4 | 6.3 | 10.9 | 10.4 | 56.2±28.5 |
| hopp-m | expert | 79.9 | 83.7 | **87.4** | 35.0 | 3.5 | 7.8 | 3.5 | 57.6±32.7 |
| hopp-m-r | medium | 36.0 | 39.4 | 35.9 | 35.5 | 11.1 | 23.0 | 16.8 | **63.1**±3.4 |
| hopp-m-r | medium-expert | 36.1 | 34.1 | 36.1 | 47.5 | 3.8 | **54.0** | 35.3 | 23.7±17.6 |
| hopp-m-r | expert | 36.0 | 36.1 | 36.1 | 49.9 | 9.8 | 19.9 | 6.7 | **62.0**±20.7 |
| hopp-m-e | medium | 66.0 | 61.1 | 65.2 | 65.3 | 21.6 | 3.4 | 14.3 | **66.8**±1.4 |
| hopp-m-e | medium-expert | 45.1 | 61.9 | **62.9** | 38.6 | 10.3 | 16.9 | 6.6 | 49.2±27.3 |
| hopp-m-e | expert | 44.9 | **84.2** | 52.8 | 29.9 | 18.7 | 10.9 | 15.5 | 68.1±16.8 |
| walk-m | medium | 34.3 | 35.2 | 41.9 | 49.6 | 31.6 | 26.9 | 27.3 | **55.7**±11.0 |
| walk-m | medium-expert | 30.2 | **51.9** | 42.3 | 43.5 | 35.8 | 19.8 | 19.1 | 37.6±8.2 |
| walk-m | expert | 56.4 | 40.7 | **60.4** | 46.7 | 35.4 | 50.2 | 38.2 | 55.7±8.0 |
| walk-m-r | medium | 11.5 | 12.5 | 22.2 | 49.7 | 17.9 | 33.7 | 6.8 | **54.2**±19.9 |
| walk-m-r | medium-expert | 9.7 | 11.2 | 7.6 | **55.9** | 24.2 | 49.8 | 28.1 | 31.3±9.7 |
| walk-m-r | expert | 7.7 | 7.4 | 7.5 | 51.9 | 18.4 | 3.1 | 18.0 | **53.7**±6.9 |
| walk-m-e | medium | 41.8 | 38.1 | 41.2 | 44.6 | 38.6 | 5.6 | **78.9** | 60.1±4.9 |
| walk-m-e | medium-expert | 22.2 | 23.6 | 28.1 | 16.5 | 15.2 | 29.2 | 33.0 | **51.4**±21.2 |
| walk-m-e | expert | 26.3 | 36.0 | 46.2 | 42.4 | 39.3 | 25.0 | 32.2 | **56.8**±11.5 |
| ant-m | medium | 50.0 | 42.3 | 54.5 | 55.4 | 31.2 | 22.5 | 17.7 | **59.2**±2.0 |
| ant-m | medium-expert | 57.8 | 54.1 | 54.5 | **60.7** | 13.0 | 7.9 | 13.5 | 60.3±5.4 |
| ant-m | expert | 59.6 | 54.2 | 49.4 | **90.4** | 7.0 | 7.0 | 11.7 | 88.7±8.9 |
| ant-m-r | medium | 43.7 | 42.0 | 41.4 | **52.8** | 31.1 | 22.4 | 30.3 | 51.7±4.6 |
| ant-m-r | medium-expert | 36.5 | 36.0 | 37.2 | 54.2 | 26.9 | 12.0 | 33.1 | **62.8**±1.9 |
| ant-m-r | expert | 24.4 | 22.1 | 24.3 | 74.7 | 27.1 | 8.9 | 25.5 | **89.9**±5.0 |
| ant-m-e | medium | 49.5 | 44.7 | 41.8 | 50.2 | 21.2 | 9.4 | 11.1 | **52.2**±4.8 |
| ant-m-e | medium-expert | 37.2 | 33.3 | 41.5 | 48.8 | 16.5 | 10.8 | 13.6 | **55.6**±3.2 |
| ant-m-e | expert | 18.7 | 17.8 | 14.4 | 78.4 | 7.2 | 8.0 | 11.7 | **88.5**±10.3 |
| Total Score | | 1193.0 | 1219.8 | 1271.0 | 1547.6 | 679.4 | 647.2 | 680 | **1900.6** |

On both morphology and kinematic shifts, DFDT consistently surpasses sequence-modelling baselines (DT, QT, DADT) with **three** exceptions and frequently outperforms strong cross-domain offline RL methods (e.g., OTDF, DARA, IGDF) across all morphology and kinematic shift tasks. Notably, DFDT achieves higher normalized scores than all baselines on **31** out of **36** tasks under the morphology shifts and **24** out of **36** tasks under the kinematic shifts. In the few settings where a competing method attains the top score, DFDT typically ranks second with a small gap, indicating broad robustness rather than narrow wins. The total normalized score improves by **63.1%** under morphology shifts and **22.8%** under kinematic shifts when using DFDT (both relative to the second-best baseline, OTDF; for reference, the gains vs. IQL are **160.5%** and **59.3%**, respectively), providing strong evidence for the method's effectiveness. Beyond these aggregates, the improvements are broad across environments (*halfcheetah, hopper, walker2d, ant*) rather than concentrated: for example, DFDT leads all **9** *halfcheetah* configurations under both morphology and kinematic shifts, and its margins are most pronounced on expert-target datasets where sequence stitching yields especially high returns. Even in rows where another method briefly tops the table, DFDT's mean typically sits within a few points while maintaining competitive seed-level stability, reinforcing that DFDT's gains reflect reliable cross-domain adaptation rather than isolated outliers.

### 5.3 TOKEN-STITCHING AND SEQUENCE STABILITY (ANSWER RQ(2))

**Token-stitching analyses setups.** To directly probe sequence semantics at stitch junctions, we precompute the junction index set $\mathcal{J}$ for every relabeled training sequence (the boundary where two fragments are concatenated, or a source $\rightarrow$ target switch). At each training checkpoint, we evaluate three quantities on a fixed validation pool of such sequences: (i) the action jump $J_a = \mathbb{E}_{t^\star \in \mathcal{J}} \|\pi(s_{t^\star}) - \pi(s_{t^\star - 1})\|_2$; (ii) the $Q$-jump $J_Q = \mathbb{E}_{t^\star \in \mathcal{J}} |Q(s_{t^\star}, \pi(s_{t^\star})) - Q(s_{t^\star - 1}, \pi(s_{t^\star - 1}))|$; and (iii) the TD residual around junctions, computed as $\mathbb{E}_{t \in \mathcal{N}(t^\star)} |r_t + \gamma V(s_{t+1}) - V(s_t)|$, where $\mathcal{N}(t^\star) = \{ t : |t - t^\star| \leq w \} \cap \{1, \ldots, T - 1\}$ denotes a small temporal neighborhood around the stitch junction index $t^\star$ with a fixed radius $w$ (in our experiments $w = 2$, i.e., two steps before or after the junction). The TD residual is then averaged over this local window to reduce single–step noise at the boundary. Curves in Fig. 2 report moving means over checkpoints for DFDT, DADT, QT, and DT under the same backbone, budget, and data.

**Results and answer to (b):** DFDT exhibits uniformly lower-level variance on all three diagnostics throughout training, indicating smoother token transitions and better local Bellman consistency at stitch points. Concretely, its action-jump mean remains $\approx 0.06 \sim 0.09$ (vs. QT rising to $0.25 \sim 0.30$ and DT/DADT $\approx 0.10 \sim 0.16$); $Q$-jumps stay near $2 \sim 3$ (vs. QT often $15 \sim 35$ and DADT spikes $> 20$); and TD residuals remain around $3 \sim 6$ (vs. QT $15 \sim 30$, DT/DADT $8 \sim 20$). Beyond lower jump means, DFDT's trajectories show markedly fewer late-training spikes, suggesting that weighted advantage conditioning and two-level filtering suppress junction value and action discontinuities as learning progresses. **These trends directly support (b):** DFDT provides stable, value-consistent sequence semantics for policy adaptation, particularly where stitching is challenging; competing sequence models exhibit larger jumps and drift, reflecting unstable semantics across stitched tokens.

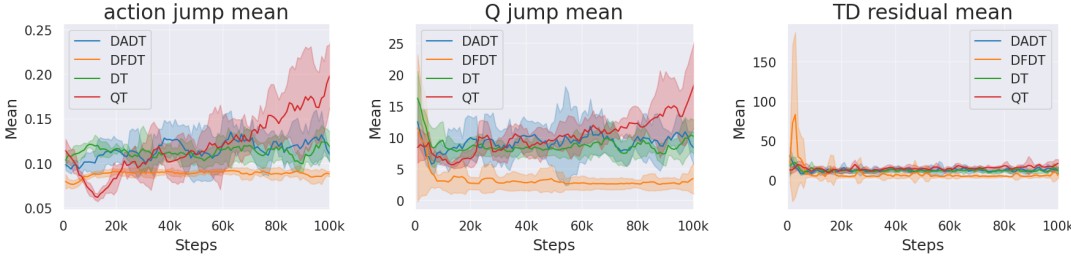

Figure 2: Mean action jump, $Q$-value jump, and TD error when evaluation.

## 6 CONCLUSION

We studied how cross-domain dynamics shifts (in rewards, horizons, and action feasibility) break DT's token-level stitchability, and proposed DFDT to explicitly restore it via two-level fragment filtering (an MMD-based state-structure gate and OT-based action-feasibility weights), a feasibility-weighted fusion distribution, advantage-conditioned tokens that replace brittle RTG with a value-consistent signal, and a lightweight $Q$-guided regularizer that suppresses junction jumps. Our analysis bounds target-domain value and performance gaps via stitchability radii and critic estimation errors, clarifying the complementary roles of MMD-based selection and OT-based weighting, and experiments across morphology, kinematic, and gravity shifts show that DFDT attains the best aggregate returns, reduces action, $Q$-value, and local TD-residual jumps, with ablations confirming the importance of both two-level filtering and advantage relabeling.

**Limitations and future work.** Our empirical evaluation focuses on simulated benchmarks such as D4RL-style cross-domain MuJoCo tasks. This choice enables controlled and repeatable comparisons under well-specified dynamics shifts, but it also means that we do not yet include real-world or larger-scale transfer experiments. In this work we instantiate DFDT with a Decision Transformer backbone for space and computational reasons, but the same two-level feasibility-weighted fusion $\mathbb{P}_{\text{mix}}^w$ is in principle compatible with TD-based offline RL methods (e.g., IQL) and actor–critic pipelines used in practice. Extending DFDT to real-world and large-scale deployments, and exploring such TD-based variants, are important directions for future work.

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

# Supplementary Material

## Table of Contents

## A   RELATED WORK

**Offline Reinforcement Learning.**   Offline RL (Yu et al., 2020) seeks to learn high-performing policies from fixed datasets without additional environment interaction, and thus must confront distributional shift and overestimation on out-of-distribution (OOD) actions (Levine et al., 2020; Kidambi et al., 2020). Constraint- or conservatism-based methods address this by penalising or avoiding unsupported actions (Lyu et al., 2022), including CQL's explicit value suppression for OOD actions (Kumar et al., 2020), IQL's advantage-weighted updates without importance sampling (Kostrikov et al., 2022), and TD3+BC's minimalist behavior-regularized regression (Fujimoto & Gu, 2021). Earlier behavior-constrained approaches, such as BCQ and BEAR, limit the learned policy's deviation from the behavior policy to reduce extrapolation error (Fujimoto et al., 2019; Kumar et al., 2019a); model-based variants (e.g., MOPO) leverage pessimistic rollouts to avoid compounding model bias (Yu et al., 2020). Benchmarks like D4RL standardise evaluation across tasks and dataset qualities, and have also catalysed analyses contrasting value-learning and supervised, return-conditioned paradigms (Fu et al., 2020; Brandfonbrener et al., 2022).

**Cross-Domain Reinforcement Learning.**   Cross-domain RL studies transfer under mismatched dynamics, morphology, sensing, or rewards, where naively pooling data across domains induces value bias and out-of-distribution actions. Early robustness strategies, domain and dynamics randomisation for sim-to-real transfer and risk-averse ensemble training, remain foundational (Tobin et al., 2017; Peng et al., 2018; Rajeswaran et al., 2017). More principled distribution alignment narrows source–target gaps by matching states and transitions via kernel MMD or optimal transport (OT) with geometry-aware costs (Gretton et al., 2012; Courty et al., 2017; Villani et al., 2008; Peyré et al., 2019). From 2024–2025, several advances clarified data selection and evaluation in the offline setting: supported cross-domain offline RL formalised the problem and constraints (Liu et al., 2024); contrastive representation learning enabled domain-aware filtering without strong labels (Wen et al., 2024); the ODRL benchmark standardised off-dynamics evaluation across gravity, morphology, and other shifts (Lyu et al., 2024); and OTDF combined OT-based filtering with dataset constraints to bound target bias (Lyu et al., 2025b). In parallel, generative augmentation synthesised stitching transitions to connect suboptimal and optimal fragments (Li et al., 2024), domain-unlabeled formulations relaxed per-transition domain tags (Nishimori et al., 2024), and sim–real co-training explored domain-invariant alignment and unbalanced OT when simulation greatly exceeds scarce

real data (Maddukuri et al., 2025; Cheng et al., 2025). Recent work such as PSEC (Liu et al., 2025) and DmC (Le Pham Van et al., 2025) also studies cross-domain offline RL. PSEC focuses on parameter-space skill expansion and composition, maintaining a library of skills and composing them in parameter space to tackle new tasks (including dynamics shifts). DmC instead uses a $k$NN-based proximity estimator and a nearest-neighbor–guided diffusion model to generate additional target-aligned trajectories under limited target data. By contrast, DFDT operates purely in the trajectory space of offline logs: it performs two-level (MMD+OT) fragment filtering and feasibility-weighted fusion, and reuses existing source trajectories without training a generative model or a reusable skill library. A full empirical comparison with PSEC and DmC under carefully matched settings is an interesting direction for future work, but is beyond the scope of the present submission due to their substantial engineering and computational requirements. Finally, sequence-modelling baselines increasingly replace brittle return-to-go conditioning with value- or advantage-aware signals, e.g., advantage-conditioned DT, critic-guided DT, and Q-regularized transformers that promote value-consistent stitching across domains (Gao et al., 2024; Wang et al., 2023; Hu et al., 2024). RADT (Wang et al., 2024) is a closely related DT-based off-dynamics method that tackles dynamics shift via *return augmentation* in return space (modifying return labels under RCSL), whereas DFDT operates in *transition space* via two-level MMD+OT fusion and advantage-conditioned tokens, making the two approaches complementary rather than overlapping.

**Conditional Sequence Modelling for Decision Making.** Viewing decision making as conditional sequence modelling enables reuse of powerful generative backbones originating from reward-/return-conditioned policies (Kumar et al., 2019b; Schmidhuber, 2019). Decision Transformer conditions on return-to-go (RTG) to autoregressively generate actions (Chen et al., 2021), Trajectory Transformer models trajectory tokens and performs planning-time search (Janner et al., 2021b), and Diffuser plans by denoising entire trajectories with diffusion models (Janner et al., 2022). However, RTG conditioning can be brittle under reward scaling or horizon mismatch and may degrade across domains; theory and empirical analyses clarify when return-conditioned supervised learning is reliable and where it fails (Brandfonbrener et al., 2022). This motivates replacing or augmenting RTG with value/advantage signals that better reflect local action quality—such as advantage-conditioned DT (Gao et al., 2024), critic-guided conditioning (Wang et al., 2023), and Q-regularized transformers (Hu et al., 2024). Building on this line, we adopt weighted advantage conditioning with Q-regularization to mitigate value jumps at stitch junctions during cross-dynamics fusion and to stabilize token-level conditioning.

## B  ADDITIONAL LEMMA B.1 AND ITS PROOF

**Assumption B.1** (Boundedness and Lipschitz). $|r_T| \leq R_{\max}$. The state value function $V$ and state-action value function $Q$ are Lipschitz in $u$ with constants $L_V, L_Q$ under the given metric $\rho$.

**Assumption B.2** (Encoder and kernel). $\|f_\phi(s)\| \leq B$, and the kernel $k$ is bounded and induces an RKHS with unit-norm ball $\{h : \|h\|_{\mathcal{H}} \leq 1\}$.

**Assumption B.3** (Compact latent image and continuity). The encoder $f_\phi : \mathcal{S} \to \mathcal{Z}$ is continuous and the latent image $K := \overline{f_\phi(\mathcal{S})} \subset \mathbb{R}^d$ is compact (e.g., via normalisation/clipping).

**Assumption B.4** (Universal kernel on $K$). The bounded kernel $k$ is universal on $K$, i.e., the induced RKHS $\mathcal{H}$ is dense in $C(K)$ with respect to the uniform norm, where $C(K)$ is the set of all real-valued continuous functions on $K$.

**Definition B.1** (Pushforward measure). Let $(X, \Sigma_X)$ and $(Y, \Sigma_Y)$ be measurable spaces, $T : X \to Y$ measurable, and $\mu$ a measure on $(X, \Sigma_X)$. The pushforward of $\mu$ by $T$, denoted $T_\#\mu$, is the measure on $(Y, \Sigma_Y)$ defined by

$$(T_\#\mu)(B) = \mu\big(T^{-1}(B)\big) \quad \forall B \in \Sigma_Y,$$

equivalently, for integrable $g : Y \to \mathbb{R}$,

$$\int_Y g \, d(T_\#\mu) = \int_X g \circ T \, d\mu.$$

**Definition B.2** (Kantorovich–Rubinstein Duality). Let $(\mathcal{X}, d)$ be a metric space and let $\mu, \nu$ be probability measures on $\mathcal{X}$ with finite first moments. The 1-Wasserstein distance is defined by the

optimal transport ("primal") problem

$$W_1(\mu, \nu) := \inf_{\pi \in \Pi(\mu,\nu)} \int_{\mathcal{X} \times \mathcal{X}} d(x,y) \, \mathrm{d}\pi(x,y),$$

where $\Pi(\mu, \nu)$ is the set of all couplings of $\mu$ and $\nu$. The Kantorovich–Rubinstein (KR) duality states that

$$W_1(\mu, \nu) = \sup_{\|f\|_{\mathrm{Lip}} \leq 1} \left\{ \int_{\mathcal{X}} f \, \mathrm{d}\mu - \int_{\mathcal{X}} f \, \mathrm{d}\nu \right\}, \qquad \|f\|_{\mathrm{Lip}} := \sup_{x \neq y} \frac{|f(x) - f(y)|}{d(x,y)}.$$

A directly usable inequality derived from the Kantorovich–Rubinstein Duality can be expressed as follows: For any $L$-Lipschitz function $g : \mathcal{X} \to \mathbb{R}$,

$$\left| \mathbb{E}_\mu[g] - \mathbb{E}_\nu[g] \right| \leq L \, W_1(\mu, \nu).$$

In particular, if $g$ is 1-Lipschitz, then

$$\left| \mathbb{E}_\mu[g] - \mathbb{E}_\nu[g] \right| \leq W_1(\mu, \nu).$$

**Lemma B.1** (Expectation deviation under the weighted data fusion). *For any 1-Lipschitz $g(u)$ and any $h$ with $\|h\|_{\mathcal{H}} \leq 1$, we have $\left| \mathbb{E}_{\mathbb{P}_{\mathrm{mix}}^w} g - \mathbb{E}_{\mathbb{P}_T} g \right| \leq \beta \, \Delta_w$ and $\left| \mathbb{E}_{\mathbb{P}_{\mathrm{mix}}^w} h - \mathbb{E}_{\mathbb{P}_T} h \right| \leq \beta \, \Delta_m$.*

*Proof.* Let $\pi_z : \mathcal{U} \to \mathcal{Z}$ map $u = (s, a, s')$ to the latent state $z = f_\phi(s)$. Denote the pushforward marginals by $\mu_T := \pi_{z\#} \mathbb{P}_T$ and $\mu_S^w := \pi_{z\#} \mathbb{P}_S^w$. For each retained fragment $\tau^S$ (i.e., $I_m(\tau^S) = 1$), let $\mu_\tau$ be its latent-state (empirical or normalized) distribution.

**Step 1 (Lipschitz part via Kantorovich–Rubinstein duality).** By linearity of expectation under the convex mixture, we have

$$\mathbb{E}_{\mathbb{P}_{\mathrm{mix}}^w} g = (1 - \beta) \, \mathbb{E}_{\mathbb{P}_T} g + \beta \, \mathbb{E}_{\mathbb{P}_S^w} g \Rightarrow \mathbb{E}_{\mathbb{P}_{\mathrm{mix}}^w} g - \mathbb{E}_{\mathbb{P}_T} g = \beta \left( \mathbb{E}_{\mathbb{P}_S^w} g - \mathbb{E}_{\mathbb{P}_T} g \right).$$

Taking absolute values and applying the Kantorovich–Rubinstein duality on $(\mathcal{U}, \rho)$,

$$\sup_{\mathrm{Lip}(g) \leq 1} \left| \mathbb{E}_{\mathbb{P}_S^w} g - \mathbb{E}_{\mathbb{P}_T} g \right| = W_1(\mathbb{P}_S^w, \mathbb{P}_T) = \Delta_w.$$

Hence, for any 1-Lipschitz $g$,

$$\left| \mathbb{E}_{\mathbb{P}_{\mathrm{mix}}^w} g - \mathbb{E}_{\mathbb{P}_T} g \right| \leq \beta \, \Delta_w.$$

**Step 2 (MMD part on latent states).** For $h \in \mathcal{H}$ acting on $z$, expectations under triple distributions reduce to those under their latent pushforwards: $\mathbb{E}_{\mathbb{P}} h := \mathbb{E}_{z \sim \pi_{z\#} \mathbb{P}}[h(z)]$. As above,

$$\mathbb{E}_{\mathbb{P}_{\mathrm{mix}}^w} h - \mathbb{E}_{\mathbb{P}_T} h = \beta \left( \mathbb{E}_{\mathbb{P}_S^w} h - \mathbb{E}_{\mathbb{P}_T} h \right) = \beta \left( \mathbb{E}_{\mu_S^w} h - \mathbb{E}_{\mu_T} h \right).$$

Taking the supremum over the unit RKHS ball and using the kernel mean embedding characterisation of MMD,

$$\sup_{\|h\|_{\mathcal{H}} \leq 1} \left| \mathbb{E}_{\mu_S^w} h - \mathbb{E}_{\mu_T} h \right| = \mathrm{MMD}_k(\mu_S^w, \mu_T).$$

Therefore, for any $\|h\|_{\mathcal{H}} \leq 1$,

$$\left| \mathbb{E}_{\mathbb{P}_{\mathrm{mix}}^w} h - \mathbb{E}_{\mathbb{P}_T} h \right| \leq \beta \, \mathrm{MMD}_k(\mu_S^w, \mu_T).$$

**Step 3 (Bounding $\mathrm{MMD}_k(\mu_S^w, \mu_T)$ by $\Delta_m$).** Since $\mathbb{P}_S^w$ places mass only on retained fragments, its latent marginal is a convex combination $\mu_S^w = \sum_{\tau^S : I_m(\tau^S) = 1} \alpha_\tau \mu_\tau$ with $\alpha_\tau \geq 0$, $\sum_\tau \alpha_\tau = 1$. As an IPM, MMD is convex in its first argument; thus,

$$\mathrm{MMD}_k \left( \sum_\tau \alpha_\tau \mu_\tau, \; \mu_T \right) \leq \sum_\tau \alpha_\tau \, \mathrm{MMD}_k(\mu_\tau, \mu_T) \leq \sup_{\tau : I_m(\tau) = 1} \mathrm{MMD}_k(\mu_\tau, \mu_T) = \Delta_m.$$

Hence $\mathrm{MMD}_k(\mu_S^w, \mu_T) \leq \Delta_m$, and therefore

$$\left| \mathbb{E}_{\mathbb{P}_{\mathrm{mix}}^w} h - \mathbb{E}_{\mathbb{P}_T} h \right| \leq \beta \, \Delta_m, \qquad \forall \|h\|_{\mathcal{H}} \leq 1.$$

Combining the Lipschitz/Wasserstein bound (Step 1) and the RKHS/MMD bound (Steps 2–3) yields

$$\left| \mathbb{E}_{\mathbb{P}_{\mathrm{mix}}^w} g - \mathbb{E}_{\mathbb{P}_T} g \right| \leq \beta \, \Delta_w, \qquad \left| \mathbb{E}_{\mathbb{P}_{\mathrm{mix}}^w} h - \mathbb{E}_{\mathbb{P}_T} h \right| \leq \beta \, \Delta_m,$$

as claimed. $\qquad \square$

## C ADDITIONAL LEMMA C.2 AND ITS PROOF

**Definition C.1** (Polish space). A topological space $(X, \tau)$ is called *Polish* if it is

- **separable**: there exists a countable dense subset $D \subseteq X$, and

- **completely metrizable**: there exists a metric $d$ that generates $\tau$ and under which $(X, d)$ is complete.

Equivalently, a Polish space is a separable, complete metric space (up to homeomorphism). Typical examples: $\mathbb{R}^n$ with the Euclidean topology, any closed subset of a Polish space, and countable products of Polish spaces.

**Definition C.2** (Quotient map and induced map). Let $f_\phi : \mathcal{S} \to \mathcal{Z}$ be a continuous map. Define an equivalence relation on $\mathcal{S}$ by $s \sim \tilde{s} \iff f_\phi(s) = f_\phi(\tilde{s})$, and let $q : \mathcal{S} \to \mathcal{S}/\sim$ be the canonical quotient map $q(s) = [s]$. Write $K := f_\phi(\mathcal{S}) \subseteq \mathcal{Z}$ for the image (with the subspace topology).

There is a unique map

$$\bar{f} : \mathcal{S}/\sim \longrightarrow K, \qquad \bar{f}([s]) = f_\phi(s),$$

such that $f_\phi = \bar{f} \circ q$. The map $\bar{f}$ is a bijection. Equipping $\mathcal{S}/\sim$ with the quotient topology (induced by $q$), $\bar{f}$ is a homeomorphism *iff* $f_\phi$ is a quotient map onto $K$ (equivalently, the subspace topology on $K$ agrees with the quotient topology via $f_\phi$). In this case, we (canonically) identify $K$ with the quotient $\mathcal{S}/\sim$ and call $q$ the quotient map associated with $f_\phi$.

**Lemma C.1** (Continuity of the quotient map and induced factor). *Let $(\mathcal{S}, \tau_\mathcal{S})$ be a topological space and $\sim$ an equivalence relation on $\mathcal{S}$. Equip $\mathcal{S}/\sim$ with the quotient topology*

$$U \subseteq \mathcal{S}/\sim \text{ is open} \iff q^{-1}(U) \in \tau_\mathcal{S},$$

*where $q : \mathcal{S} \to \mathcal{S}/\sim$, $q(s) = [s]$, is the canonical projection. Then $q$ is continuous.*

*Moreover, let $f_\phi : \mathcal{S} \to \mathcal{Z}$ be continuous and define $s \sim \tilde{s} \iff f_\phi(s) = f_\phi(\tilde{s})$. Writing $K := f_\phi(\mathcal{S}) \subseteq \mathcal{Z}$ with the subspace topology, there exists a unique map*

$$\bar{f} : \mathcal{S}/\sim \longrightarrow K, \qquad \bar{f}([s]) = f_\phi(s),$$

*such that $f_\phi = \bar{f} \circ q$, and $\bar{f}$ is continuous.*

*Proof.* By the definition of the quotient topology, for every open $U \subseteq \mathcal{S}/\sim$ we have $q^{-1}(U) \in \tau_\mathcal{S}$, hence $q$ is continuous.

For the second part, the definition of $\bar{f}$ is well-posed because $s \sim \tilde{s}$ implies $f_\phi(s) = f_\phi(\tilde{s})$. Uniqueness follows from $f_\phi = \bar{f} \circ q$. To prove continuity of $\bar{f}$, let $O \subseteq K$ be open (in the subspace topology). Then

$$q^{-1}\big(\bar{f}^{-1}(O)\big) = \{\, s \in \mathcal{S} : \bar{f}(q(s)) \in O \,\} = \{\, s \in \mathcal{S} : f_\phi(s) \in O \,\} = f_\phi^{-1}(O),$$

which is open in $\mathcal{S}$ since $f_\phi$ is continuous. By the quotient definition, this implies $\bar{f}^{-1}(O)$ is open in $\mathcal{S}/\sim$, i.e., $\bar{f}$ is continuous. $\qquad\square$

**Lemma C.2** (Approximate value-in-RKHS from universality and latent sufficiency). *Under Assumptions B.1, B.2, 4.1, B.3, B.4, and 4.2, for every $\eta > 0$ there exists $h_V \in \mathcal{H}$ such that*

$$\sup_{s \in \mathcal{S}} |V(s) - h_V(f_\phi(s))| \leq \varepsilon_H + \eta.$$

*Proof.* By Assumption B.1, $V$ is Lipschitz in $u = (s, a, s')$, hence in $s$; thus $V$ is continuous on $\mathcal{S}$. By Assumption B.3, $f_\phi$ is continuous and $K = \overline{f_\phi(\mathcal{S})}$ is compact; hence the quotient map induces a continuous function on $K$ up to the fibre variation. Define a (measurable) section $\sigma : K \to \mathcal{S}$ with $f_\phi(\sigma(z)) = z$ (e.g., choose any representative in each fibre) and set $\widetilde{V}(z) := V(\sigma(z))$. For any $s \in \mathcal{S}$ with $z = f_\phi(s)$,

$$|V(s) - \widetilde{V}(f_\phi(s))| = |V(s) - V(\sigma(z))| \leq \varepsilon_H$$

by Assumption 4.2. Therefore,

$$\sup_{s \in \mathcal{S}} |V(s) - \widetilde{V}(f_\phi(s))| \le \varepsilon_H.$$

Now $\widetilde{V} \in C(K)$ because $V$ and $f_\phi$ are continuous and $K$ is compact, where $C(K)$ is the set of all real-valued continuous functions on $K$. By universality (Assumption B.4), for any $\eta > 0$ there exists $h_V \in \mathcal{H}$ such that $\sup_{z \in K} |\widetilde{V}(z) - h_V(z)| \le \eta$. Combining the two displays gives

$$\sup_{s \in \mathcal{S}} |V(s) - h_V(f_\phi(s))| \le \sup_s |V(s) - \widetilde{V}(f_\phi(s))| + \sup_{z \in K} |\widetilde{V}(z) - h_V(z)| \le \varepsilon_H + \eta,$$

as claimed. The RKHS norm bound $\|h_V\|_{\mathcal{H}} \le C_V(\eta)$ follows from standard RKHS approximation estimates and can be absorbed into constants elsewhere. □

*Remark* C.1. If $f_\phi$ is *value-sufficient* (i.e., $V(s) = \widetilde{V}(f_\phi(s))$ up to a small error), then Assumption 4.2 holds with $\varepsilon_H = 0$. In practice, $\varepsilon_H$ can be made small by training $f_\phi$ to preserve value-relevant information (e.g., adding an auxiliary head $s \mapsto V(s)$ or using value-aware representation-learning objectives) and by normalizing the latent $K$ to be compact. In partially observed domains, the same encoder can be applied to observation histories, i.e., $z_t = f_\phi(o_{0:t})$, so DFDT operates on a latent state representation while the value-sufficiency requirement is imposed on $z_t$ rather than raw observations.

## D  ADDITIONAL LEMMA D.1 AND ITS PROOF

**Lemma D.1** (Weighted Bellman error transfer). *Let* $y = r(s, a) + \gamma V(s')$. *There exist constants* $R_1, R_2, R_3 > 0$ *(depending on* $R_{\max}, L_V$ *and the encoder/kernel bounds) such that*

$$\left| \mathbb{E}_{\mathbb{P}_{\text{mix}}^w}[y - V(s)] - \mathbb{E}_{\mathbb{P}_T}[y - V(s)] \right| \le \beta (R_1 \Delta_m + R_2 \Delta_w) + 2\beta \varepsilon_H, \qquad (14)$$

$$\left| \mathbb{E}_{\mathbb{P}_{\text{mix}}^w}[y - Q(s, a)] - \mathbb{E}_{\mathbb{P}_T}[y - Q(s, a)] \right| \le \beta (R_1 \Delta_m + R_3 \Delta_w) + 2\beta \varepsilon_H. \qquad (15)$$

*Proof.* **Step 1 (Bounding the** $V(s)$ **term).** Write the one-step TD residual as

$$R(u) = y - V(s) = r_T(s, a) + \gamma V(s') - V(s), \qquad u = (s, a, s').$$

By linearity of expectation under the mixture,

$$\mathbb{E}_{\mathbb{P}_{\text{mix}}^w} R - \mathbb{E}_{\mathbb{P}_T} R = \beta \big( \mathbb{E}_{\mathbb{P}_S^w} R - \mathbb{E}_{\mathbb{P}_T} R \big).$$

Hence

$$\left| \mathbb{E}_{\mathbb{P}_{\text{mix}}^w} R - \mathbb{E}_{\mathbb{P}_T} R \right| \le \beta \Big( \underbrace{\left| \mathbb{E}_{\mathbb{P}_S^w} y - \mathbb{E}_{\mathbb{P}_T} y \right|}_{(\mathrm{I})} + \underbrace{\left| \mathbb{E}_{\mathbb{P}_S^w} V(s) - \mathbb{E}_{\mathbb{P}_T} V(s) \right|}_{\mathrm{II}} \Big). \qquad (16)$$

*Term I: Bounding the* $(r_T + \gamma V)$ *term via the* $W_1$ *distance.* According to Assumption B.1, $r_T$ is $L_r$-Lipschitz in $(s, a)$ (or bounded by $R_{\max}$ and $L_r$ finite) and $V$ is $L_V$-Lipschitz in $s'$ under the given metric $\rho$ on triples $u = (s, a, s')$. Then the function

$$f(u) = r_T(s, a) + \gamma V(s')$$

is $L_f$-Lipschitz with $L_f \le L_r + \gamma L_V$. By the Kantorovich–Rubinstein duality,

$$\left| \mathbb{E}_{\mathbb{P}_S^w} f - \mathbb{E}_{\mathbb{P}_T} f \right| \le L_f W_1(\mathbb{P}_S^w, \mathbb{P}_T) = L_f \Delta_w.$$

Absorb $L_g$ into a constant $R_2 > 0$ to obtain

$$\left| \mathbb{E}_{\mathbb{P}_S^w}[r_T(s, a) + \gamma V(s')] - \mathbb{E}_{\mathbb{P}_T}[r_T(s, a) + \gamma V(s')] \right| \le R_2 \Delta_w. \qquad (17)$$

*Term II: Bounding the* $V(s)$ *term via MMD.* Let $z = f_\phi(s)$ be the latent state and let $\pi_z(u) = z$. Denote the latent pushforwards $\mu_T = \pi_{z\#}\mathbb{P}_T$ and $\mu_S^w = \pi_{z\#}\mathbb{P}_S^w$. Then by Lemma B.1 and C.2

$$\begin{aligned}
\left| \mathbb{E}_{\mathbb{P}_S^w} V(s) - \mathbb{E}_{\mathbb{P}_T} V(s) \right| &\le \left| \mathbb{E}_{z \sim \mu_S^w} h_V(z) - \mathbb{E}_{z \sim \mu_T} h_V(z) \right| + 2\varepsilon_H \\
&\le \|h_V\|_{\mathcal{H}} \, \mathrm{MMD}_k(\mu_S^w, \mu_T) + 2\varepsilon_H \\
&\le C_V \, \mathrm{MMD}_k(\mu_S^w, \mu_T) + 2\varepsilon_H.
\end{aligned}$$

Since $\mathbb{P}_S^w$ is supported on retained fragments and $\mu_S^w$ is their convex combination, MMD convexity yields $\mathrm{MMD}_k(\mu_S^w, \mu_T) \le \Delta_m$, hence

$$\left| \mathbb{E}_{\mathbb{P}_S^w} V(s) - \mathbb{E}_{\mathbb{P}_T} V(s) \right| \le C_V \Delta_m + 2\varepsilon_H. \tag{18}$$

Let $R_1 := C_V$ and plug Eq. (17) and Eq. (18) into Eq. (16) gives

$$\left| \mathbb{E}_{\mathbb{P}_{\mathrm{mix}}^w} R - \mathbb{E}_{\mathbb{P}_T} R \right| \le \beta \left( R_1 \Delta_m + R_2 \Delta_w + 2\varepsilon_H \right), \tag{19}$$

which is the desired bound about $V$.

**Step 2 (Bounding the $Q(s,a)$ term).** We want to bound the distributional shift of the $Q$–residual $y(u) - Q(s,a)$ between $\mathbb{P}_{\mathrm{mix}}^w$ and $\mathbb{P}_T$. Introduce and subtract $V(s)$:

$$y - Q = \underbrace{(y - V(s))}_{\text{(I)}} + \underbrace{(V(s) - Q(s,a))}_{\text{(II)}}.$$

Hence, by the triangle inequality,

$$\left| \mathbb{E}_{\mathbb{P}_{\mathrm{mix}}^w}[y - Q] - \mathbb{E}_{\mathbb{P}_T}[y - Q] \right| \le \underbrace{\left| \mathbb{E}_{\mathbb{P}_{\mathrm{mix}}^w}[y - V(s)] - \mathbb{E}_{\mathbb{P}_T}[y - V(s)] \right|}_{\text{Term (I)}}$$

$$+ \underbrace{\left| \mathbb{E}_{\mathbb{P}_{\mathrm{mix}}^w}[V(s) - Q(s,a)] - \mathbb{E}_{\mathbb{P}_T}[V(s) - Q(s,a)] \right|}_{\text{Term (II)}}.$$

*Term (I).* This is exactly the value–residual transfer term handled in Step 1.

*Term (II).* Let $g(u) := V(s) - Q(s,a)$. According to Assumption B.1, $g$ is Lipschitz in $u$ under the metric $\rho$, so that $\mathrm{Lip}(g) \le L_V + L_Q$. By the Kantorovich–Rubinstein duality,

$$\left| \mathbb{E}_{\mathbb{P}_{\mathrm{mix}}^w}[g] - \mathbb{E}_{\mathbb{P}_T}[g] \right| \le \mathrm{Lip}(g) \, W_1(\mathbb{P}_{\mathrm{mix}}^w, \mathbb{P}_T) = \beta \, \widetilde{R}_2 \, \Delta_w, \tag{II}$$

with $\widetilde{R}_2 := L_V + L_Q$ absorbed into constants.

Combining (I) and (II) and taking a supremum over $(s,a)$ (or dropping the conditioning) yields

$$\left| \mathbb{E}_{\mathbb{P}_{\mathrm{mix}}^w}[y - Q] - \mathbb{E}_{\mathbb{P}_T}[y - Q] \right| \le \beta \left( R_1 \Delta_m + (R_2 + \widetilde{R}_2) \Delta_w \right) + 2\beta \, \varepsilon_H.$$

Renaming $R_3 \leftarrow R_2 + \widetilde{R}_2$ gives the stated form

$$\left| \mathbb{E}_{\mathbb{P}_{\mathrm{mix}}^w}[y - Q(s,a)] - \mathbb{E}_{\mathbb{P}_T}[y - Q(s,a)] \right| \le \beta \left( R_1 \Delta_m + R_3 \Delta_w \right) + 2\beta \, \varepsilon_H.$$

$\square$

**Corollary D.1** (Weighted Bellman transfer for $\pi_{\mathrm{mix}}$). *Let $y = r_T(s,a) + \gamma V(s')$ as in Lemma D.1. For any measurable $\varphi : \mathcal{S} \to \mathbb{R}$ with $\|\varphi\|_\infty \le 1$, there exist constants $R_1, R_2 > 0$ (depending only on $R_{\max}$, $L_V$ and the encoder/kernel bounds, absorbed into the same symbols) such that*

$$\left| \mathbb{E}_{(s,a,s') \sim \mathbb{P}_T^{\pi_{\mathrm{mix}}}}[\varphi(s) \, y] - \mathbb{E}_{(s,a,s') \sim \mathbb{P}_{\mathrm{mix}}^w}[\varphi(s) \, y] \right| \le \beta \left( R_1 \Delta_m + R_2 \Delta_w \right) + 2\beta \, \varepsilon_H.$$

Proof sketch. *Apply Lemma D.1 to $\psi(s,a,s') := \varphi(s) \, y(s,a,s')$ (bounded by $\|\varphi\|_\infty \le 1$) and note that the target-side law is $\mathbb{P}_T^{\pi_{\mathrm{mix}}}$.* $\square$

# E  PROOF OF THEOREM 4.1

**Definition E.1** (Occupancy–weighted $L_1$ norms). For any policy $\pi$, let $d_T^\pi$ be the normalized discounted state–occupancy on $\mathcal{S}$ and $d_T^\pi \otimes \pi$ the corresponding state–action occupancy on $\mathcal{S} \times \mathcal{A}$. Define

$$\|f\|_{1,d_T^\pi} := \mathbb{E}_{s \sim d_T^\pi}[\,|f(s)|\,], \qquad \|g\|_{1,d_T^\pi \otimes \pi} := \mathbb{E}_{\substack{s \sim d_T^\pi \\ a \sim \pi(\cdot|s)}}[\,|g(s,a)|\,].$$

**Definition E.2** (Radon–Nikodym derivative). Let $(\Omega, \mathcal{F})$ be a measurable space and let $\nu, \mu$ be $\sigma$-finite measures with $\nu \ll \mu$ (i.e., $\mu(A) = 0 \Rightarrow \nu(A) = 0$ for all $A \in \mathcal{F}$). The *Radon–Nikodym derivative* of $\nu$ with respect to $\mu$ is the (a.e.-unique) measurable function $\frac{d\nu}{d\mu} : \Omega \to [0, \infty]$ such that

$$\nu(A) = \int_A \frac{d\nu}{d\mu} \, d\mu \qquad \text{for all } A \in \mathcal{F}.$$

Equivalently, for any $\mu$-integrable $g$,

$$\int g \, d\nu = \int g \, \frac{d\nu}{d\mu} \, d\mu.$$

**Lemma E.1** (Radon–Nikodym). *If $\nu, \mu$ are $\sigma$-finite and $\nu \ll \mu$, then the derivative $\frac{d\nu}{d\mu}$ exists and is unique $\mu$-almost everywhere.*

**Lemma E.2** (Performance–difference lemma (discounted MDP) Schulman et al. (2015); Kakade & Langford (2002)). *Consider a discounted MDP $\mathcal{M} = (\mathcal{S}, \mathcal{A}, P, r, \gamma)$ with $\gamma \in [0, 1)$ and an initial-state distribution $\rho$ on $\mathcal{S}$. For any policies $\pi, \pi'$, define*

$$V^\pi(s) := \mathbb{E}\Big[\sum_{t \geq 0} \gamma^t r(s_t, a_t) \Big| s_0 = s, \ a_t \sim \pi(\cdot|s_t), \ s_{t+1} \sim P(\cdot|s_t, a_t)\Big],$$

$$Q^\pi(s, a) := r(s, a) + \gamma \, \mathbb{E}[V^\pi(s') \mid s, a],$$

*and the advantage $A^\pi(s, a) := Q^\pi(s, a) - V^\pi(s)$. Let the (discounted) performance be $J(\pi) := \mathbb{E}_{s_0 \sim \rho}[V^\pi(s_0)]$. Define the normalized discounted state-occupancy of $\pi'$:*

$$d_\rho^{\pi'}(s) := (1 - \gamma) \sum_{t=0}^\infty \gamma^t \, \Pr(s_t = s \mid s_0 \sim \rho, \ \pi').$$

*Then*

$$J(\pi') - J(\pi) = \frac{1}{1 - \gamma} \, \mathbb{E}_{s \sim d_\rho^{\pi'}, \ a \sim \pi'(\cdot|s)}\big[A^\pi(s, a)\big].$$

*In particular, if $\pi'$ is deterministic, this reduces to*

$$J(\pi') - J(\pi) = \frac{1}{1 - \gamma} \, \mathbb{E}_{s \sim d_\rho^{\pi'}}\big[A^\pi(s, \pi'(s))\big].$$

**Lemma E.3** (Bellman operator is a $\gamma$-contraction). *Let $\mathcal{B}(\mathcal{S})$ denote the bounded real-valued functions on $\mathcal{S}$ equipped with the norm $\|\cdot\|$. Fix $\gamma \in [0, 1)$ and a target-domain Markov kernel $P(s' \mid s, a)$. Let $\pi(a \mid s)$ be any (possibly stochastic) conditional law of actions given $s$. Define the target Bellman operator*

$$(\mathcal{T}f)(s) := \mathbb{E}_{\substack{a \sim \pi(\cdot|s) \\ s' \sim P(\cdot|s,a)}} \big[r_T(s, a) + \gamma f(s')\big].$$

*Then, for all $f, g \in \mathcal{B}(\mathcal{S})$,*

$$\|\mathcal{T}f - \mathcal{T}g\| \leq \gamma \|f - g\|.$$

*Consequently, $\mathcal{T}$ is a $\gamma$-contraction and has a unique fixed point $V^* \in \mathcal{B}(\mathcal{S})$.*

*Proof.* For any $s \in \mathcal{S}$,

$$(\mathcal{T}f - \mathcal{T}g)(s) = \gamma \, \mathbb{E}[f(s') - g(s') \mid s].$$

Hence $\big|(\mathcal{T}f - \mathcal{T}g)(s)\big| \leq \gamma \, \|f - g\|$, and taking the supremum over $s$ gives $\|\mathcal{T}f - \mathcal{T}g\| \leq \gamma \|f - g\|$. By the Banach fixed-point theorem, $\mathcal{T}$ admits a unique fixed point $V^*$. $\qquad\square$

**Assumption E.1** (Occupancy-to-sampling concentrability). There exists a constant $\chi \in (0, \infty)$ such that for every stationary policy $\pi$, the normalized discounted occupancy satisfies $d_T^\pi \ll \mathbb{P}_T$ and

$$\left\|\frac{d d_T^\pi}{d \mathbb{P}_T}\right\|_\infty \leq \chi.$$

All such density-ratio constants are absorbed into numerical constants below.

**Lemma E.4** (Policy mismatch for evaluation residuals). *Let $B_V := R_{\max} + \gamma\|V\|_\infty$. Then for any two stationary policies $\pi_1, \pi_2$,*

$$\|\mathcal{T}_T^{\pi_1} V - \mathcal{T}_T^{\pi_2} V\|_{1,\mathbb{P}_T} \ \leq \ B_V \cdot \mathbb{E}_{s\sim\mathbb{P}_T}\big[\|\pi_1(\cdot|s) - \pi_2(\cdot|s)\|_1\big].$$

*In particular, with $\Delta_\pi := \mathbb{E}_{s\sim\mathbb{P}_T}\big[\|\pi_T^*(\cdot|s) - \pi_{\mathrm{mix}}(\cdot|s)\|_1\big]$,*

$$\|\mathcal{T}_T^{\pi_T^*} V - \mathcal{T}_T^{\pi_{\mathrm{mix}}} V\|_{1,\mathbb{P}_T} \ \leq \ B_V\,\Delta_\pi.$$

Proof. *For each $s$, $|\mathbb{E}_{a\sim\pi_1}[g_s(a)] - \mathbb{E}_{a\sim\pi_2}[g_s(a)]| \leq \|g_s\|_\infty \|\pi_1(\cdot|s) - \pi_2(\cdot|s)\|_1$, where $g_s(a) := \mathbb{E}[r_T(s,a) + \gamma V(s') \,|\, s, a]$ satisfies $\|g_s\|_\infty \leq B_V$. Average over $s \sim \mathbb{P}_T$.* $\qquad\square$

**Theorem 4.1** (Performance bound under stitchability radii). *Under Assumptions 4.1–4.3, training with $\mathbb{P}_{\mathrm{mix}}^w$ yields estimators $V$ and $Q$. Let $V_T$ and $Q_T$ be the state and state–action value functions learned from the target dataset. Let $\pi_T^*$ and $\pi_{\mathrm{mix}}$ denote any optimal policies learned from the target MDP $\mathbb{P}_T$ and the mixed MDP $\mathbb{P}_{\mathrm{mix}}^w$, respectively. Let $d_T^* \otimes \pi_T^*$ be the normalized discounted state–action occupancy of $\pi_T^*$ under $\mathbb{P}_T$ and $\Delta_\pi := \mathbb{E}_{s\sim\mathbb{P}_T}\big[\|\pi_T^*(\cdot|s) - \pi_{\mathrm{mix}}(\cdot|s)\|_1\big]$. Then, for some constants $C_1, C_2, C_3, C_H, C_\pi > 0$,*

$$\|V - V_T\|_{1,\mathbb{P}_T} \ \leq \ \frac{C_1\,\beta\,(\Delta_m + \Delta_w) + 2\beta\,\varepsilon_H + \varepsilon_V}{1 - \gamma}, \tag{11}$$

$$\|Q - Q_T\|_{1,\mathbb{P}_T} \ \leq \ \frac{C_2\,\beta\,(\Delta_m + \Delta_w) + 2\beta\,\varepsilon_H + \varepsilon_Q}{1 - \gamma}. \tag{12}$$

*Moreover, by a performance-difference bound,*

$$J_T(\pi_T^*) - J_T(\pi_{\mathrm{mix}}) \ \leq \ \frac{C_3(1 + \kappa)}{(1 - \gamma)^2}\left(\beta(\Delta_m + \Delta_w) + C_H\,\beta\,\varepsilon_H + \varepsilon_V\right) + \frac{C_\pi}{(1 - \gamma)^2}\,\Delta_\pi. \tag{13}$$

*Proof.* **Step 1 (One-step residual under the target domain).** Define the (conditional) TD residuals

$$\delta_V(s) \ := \ \mathbb{E}_{\mathbb{P}_T}\big[r_T(s,a) + \gamma V(s') - V(s) \,\big|\, s\big], \quad \delta_Q(s,a) \ := \ \mathbb{E}_{\mathbb{P}_T}\big[r_T(s,a) + \gamma V(s') - Q(s,a) \,\big|\, s, a\big].$$

Add and subtract the mixed-distribution residuals and apply the triangle inequality:

$$\|\delta_V\|_{1,\mathbb{P}_T} \ \leq \ \underbrace{\Big|\mathbb{E}_{\mathbb{P}_{\mathrm{mix}}^w}[y - V(s)]\Big|}_{\leq\,\varepsilon_V \text{ by Assumption 4.1}} + \Big|\mathbb{E}_{\mathbb{P}_T}[y - V(s)] - \mathbb{E}_{\mathbb{P}_{\mathrm{mix}}^w}[y - V(s)]\Big|,$$

where $y = r_T(s,a) + \gamma V(s')$. Taking $\sup_s$ and using that conditional deviations are bounded by unconditional ones,

$$\|\delta_V\|_{1,\mathbb{P}_T} \ \leq \ \varepsilon_V + \Big|\mathbb{E}_{\mathbb{P}_T}[y - V(s)] - \mathbb{E}_{\mathbb{P}_{\mathrm{mix}}^w}[y - V(s)]\Big|. \tag{20}$$

An identical argument yields

$$\|\delta_Q\|_{1,\mathbb{P}_T} \ \leq \ \varepsilon_Q + \Big|\mathbb{E}_{\mathbb{P}_T}[y - Q(s,a)] - \mathbb{E}_{\mathbb{P}_{\mathrm{mix}}^w}[y - Q(s,a)]\Big|. \tag{21}$$

**Step 2 (Stitchability transfer).** By Lemma D.1, there exist constants $R_1, R_2 > 0$ such that

$$\Big|\mathbb{E}_{\mathbb{P}_{\mathrm{mix}}^w}[y - V(s)] - \mathbb{E}_{\mathbb{P}_T}[y - V(s)]\Big| \ \leq \ \beta\,(R_1\,\Delta_m + R_2\,\Delta_w) + 2\beta\,\varepsilon_H, \tag{22}$$

and

$$\Big|\mathbb{E}_{\mathbb{P}_{\mathrm{mix}}^w}[y - Q(s,a)] - \mathbb{E}_{\mathbb{P}_T}[y - Q(s,a)]\Big| \ \leq \ \beta\,(R_1\,\Delta_m + R_3\,\Delta_w) + 2\beta\,\varepsilon_H. \tag{23}$$

Plugging Eq. (22) into Eq. (20) and Eq. (23) into Eq. (21) gives

$$\|\delta_V\|_{1,\mathbb{P}_T} \leq \varepsilon_V + \beta\,(R_1\,\Delta_m + R_2\,\Delta_w) + 2\beta\,\varepsilon_H, \tag{24}$$
$$\|\delta_Q\|_{1,\mathbb{P}_T} \leq \varepsilon_Q + \beta\,(R_1\,\Delta_m + R_3\,\Delta_w) + 2\beta\,\varepsilon_H. \tag{25}$$

**Step 3 (Contraction to fixed-point errors).** Let $\mathcal{T}_V$ be the (target) Bellman operator associated with $y$, i.e., $(\mathcal{T}_V f)(s) := \mathbb{E}_{\mathbb{P}_T}[r_T(s,a) + \gamma f(s') \,|\, s]$. By Lemma E.3, $T_V$ is a $\gamma$-contraction in $\|\cdot\|_{1,\mathbb{P}_T}$ with unique fixed point $V_T$. Note that $\delta_V = \mathcal{T}_V V - V$ pointwise, hence

$$\|V - V_T\|_{1,\mathbb{P}_T} \ \leq \ \frac{\|\mathcal{T}_V V - V\|_{1,\mathbb{P}_T}}{1 - \gamma} \ = \ \frac{\|\delta_V\|_{1,\mathbb{P}_T}}{1 - \gamma}.$$

Using Eq. (24) produces the inequality in Eq. (11) with $C_1 := \max\{R_1, R_2\}$ (absorbing constants).

For $Q$, define the (evaluation) Bellman operator $(\mathcal{T}_Q f)(s, a) := \mathbb{E}_{\mathbb{P}_T}[r_T(s, a) + \gamma V(s') \mid s, a]$, for any $f, g$,

$$(\mathcal{T}_Q f)(s, a) = r_T(s, a) + \gamma \, \mathbb{E}[V(s') \mid s, a] \quad \text{does not depend on } f \text{ at all,}$$

hence $(\mathcal{T}_Q f) - (\mathcal{T}_Q g) \equiv 0$ and $\|\mathcal{T}_Q f - \mathcal{T}_Q g\|_{1, \mathbb{P}_T} = 0 \leq \gamma \|f - g\|_{1, \mathbb{P}_T}$. Therefore $\mathcal{T}_Q$ is (trivially) a $\gamma$-contraction with fixed point $Q_T$ for the target problem tied to $V$.[1] Since $\delta_Q = \mathcal{T}_Q Q - Q$, we obtain

$$\|Q - Q_T\|_{1, \mathbb{P}_T} \leq \frac{\|\delta_Q\|_{1, \mathbb{P}_T}}{1 - \gamma} \leq \frac{\varepsilon_Q + \beta(R_1 \Delta_m + R_3 \Delta_w) + 2\beta \varepsilon_H}{1 - \gamma},$$

which yields the second inequality in Eq. (12) (renaming the constant to $C_2$).

**Step 4 (Performance bound).** By the performance–difference lemma in occupancy form,

$$J_T(\pi_T^*) - J_T(\pi_{\mathrm{mix}}) \leq \frac{1}{1 - \gamma} \|V_T^* - V_T^{\mathrm{mix}}\|_{1, d_T^*} \leq \frac{1}{1 - \gamma} \Big(\|V_T^* - V\|_{1, d_T^*} + \|V - V_T^{\mathrm{mix}}\|_{1, d_T^*}\Big). \tag{26}$$

For the second term, change measure to $d_T^{\mathrm{mix}}$ by Assumption 4.3:

$$\|V - V_T^{\mathrm{mix}}\|_{1, d_T^*} \leq \kappa \|V - V_T^{\mathrm{mix}}\|_{1, d_T^{\mathrm{mix}}}, \qquad \kappa := \left\|\frac{\mathrm{d} d_T^*}{\mathrm{d} d_T^{\mathrm{mix}}}\right\|_\infty.$$

Next, relate value gaps to evaluation residuals along the corresponding policy (standard residual-to-value inequality):

$$\|V_T^\pi - V\|_{1, d_T^\pi} \leq \frac{1}{1 - \gamma} \|\mathcal{T}_T^\pi V - V\|_{1, d_T^\pi}, \qquad \pi \in \{\pi_T^*, \pi_{\mathrm{mix}}\}.$$

Therefore, from Eq. (26),

$$J_T(\pi_T^*) - J_T(\pi_{\mathrm{mix}}) \leq \frac{1}{(1 - \gamma)^2} \Big(\|\mathcal{T}_T^{\pi_T^*} V - V\|_{1, d_T^*} + \kappa \|\mathcal{T}_T^{\pi_{\mathrm{mix}}} V - V\|_{1, d_T^{\mathrm{mix}}}\Big). \tag{27}$$

Use Assumption E.1 to transfer both norms to $\mathbb{P}_T$:

$$\|\mathcal{T}_T^\pi V - V\|_{1, d_T^\pi} \leq \chi \|\mathcal{T}_T^\pi V - V\|_{1, \mathbb{P}_T}, \qquad \pi \in \{\pi_T^*, \pi_{\mathrm{mix}}\}.$$

Fix $\pi \in \{\pi_T^*, \pi_{\mathrm{mix}}\}$ and decompose (now with $m_{\mathrm{mix}}$ *defined* as the Bellman operator under $\pi_{\mathrm{mix}}$):

$$\|\mathcal{T}_T^\pi V - V\|_{1, \mathbb{P}_T} \leq \underbrace{\|\mathcal{T}_T^\pi V - \mathcal{T}_T^{\pi_{\mathrm{mix}}} V\|_{1, \mathbb{P}_T}}_{\text{policy mismatch}} + \underbrace{\|\mathcal{T}_T^{\pi_{\mathrm{mix}}} V - \mathbb{E}_{\mathbb{P}_{\mathrm{mix}}^w}[y \mid s]\|_{1, \mathbb{P}_T}}_{\text{distribution shift}} + \underbrace{\|\mathbb{E}_{\mathbb{P}_{\mathrm{mix}}^w}[y \mid s] - V(s)\|_{1, \mathbb{P}_T}}_{\text{estimation}}, \tag{28}$$

where $y = r_T(s, a) + \gamma V(s')$. The *policy-mismatch* term is bounded by Lemma E.4:

$$\|\mathcal{T}_T^\pi V - \mathcal{T}_T^{\pi_{\mathrm{mix}}} V\|_{1, \mathbb{P}_T} \leq B_V \Delta_\pi, \qquad B_V := R_{\max} + \gamma \|V\|_\infty, \ \Delta_\pi := \mathbb{E}_{s \sim \mathbb{P}_T}\big[\|\pi_T^*(\cdot|s) - \pi_{\mathrm{mix}}(\cdot|s)\|_1\big].$$

(For $\pi = \pi_{\mathrm{mix}}$ this term is 0.) The *distribution-shift* term is controlled by Corollary D.1 via the $L^1$ duality with bounded test functions:

$$\|\mathcal{T}_T^{\pi_{\mathrm{mix}}} V - \mathbb{E}_{\mathbb{P}_{\mathrm{mix}}^w}[y \mid s]\|_{1, \mathbb{P}_T} \leq \beta(R_1 \Delta_m + R_2 \Delta_w) + 2\beta \varepsilon_H.$$

For the *estimation* term, Assumption 4.1 gives an $\varepsilon_V$ bound under $\mathbb{P}_{\mathrm{mix}}^w$; changing the measuring law to $\mathbb{P}_T$ introduces only a bounded multiplicative factor (absorbed into constants). Hence, uniformly in $\pi$,

$$\|\mathcal{T}_T^\pi V - V\|_{1, \mathbb{P}_T} \leq B_V \Delta_\pi + C\Big(\beta(\Delta_m + \Delta_w) + \beta \varepsilon_H + \varepsilon_V\Big).$$

Putting this into Eq. (27) and absorbing the multiplicative constants (including $\chi$ and those from Lemma D.1) into $C_3, C_H, C_\pi$, we obtain

$$J_T(\pi_T^*) - J_T(\pi_{\mathrm{mix}}) \leq \frac{C_3(1 + \kappa)}{(1 - \gamma)^2} \Big(\beta(\Delta_m + \Delta_w) + C_H \beta \varepsilon_H + \varepsilon_V\Big) + \frac{C_\pi}{(1 - \gamma)^2} \Delta_\pi.$$

If one additionally assumes a mild *policy proximity* condition (e.g., $\Delta_\pi \leq C_s(\Delta_m + \Delta_w)$ as a stitchability consequence), the $\Delta_\pi$ term can be absorbed into the existing $\beta(\Delta_m + \Delta_w)$ term, recovering the original shape of Eq. (13).

This completes the proof. □

---

[1] Any standard control/evaluation choice can be used so long as the associated Bellman operator is a $\gamma$-contraction with a unique fixed point; the constants absorb the specific choice.

---

**Algorithm 1** DFDT Training

---

**Require:** Source domain dataset $\mathcal{D}_{\text{src}}$, target domain dataset $\mathcal{D}_{\text{tar}}$, batch size $N$, sequence length $K$, data filtering proportion $\xi$, target update rate $\eta_{\text{exp}}$

1: Initialize DT policy $\pi_\phi$, critic networks $Q_\phi$, target critic networks $Q_{\phi'}$, state value networks $V_\varphi$ and state-action value network $Q_\psi$ for computing weighted advantage-conditioned tokens, and command network $C_\omega$

2: // Offline cost computation

3: Pre-compute the MMD distance $\{d^m\}^{|\mathcal{D}_{\text{src}}|}$ using Eq. (2) and optimal transport distance $\{d^w\}^{|\mathcal{D}_{\text{src}}|}$ using Eq. (4)

4: Use the distance information $\{d^m\}^{|\mathcal{D}_{\text{src}}|}$ and $\{d^w\}^{|\mathcal{D}_{\text{src}}|}$ to augment the source dataset $\mathcal{D}_{\text{src}}$ and get $\widehat{\mathcal{D}}_{\text{src}} = \{(s_t, a_t, r_t, s'_t, \text{timesteps}, \text{masks}, d^m_t, d^w_t)\}$

5: // Compute weighted advantage-conditioned tokens

6: Train the state and state-action value networks using Eq. (6) and Eq. (5), respectively

7: Compute the advantage $A$ of each state-action pair in $\mathcal{D}_{\text{src}}$ and $\mathcal{D}_{\text{tar}}$

8: // Train the command network

9: Train the command network $C_\omega$ using the advantage information $A$ and MSE loss.

10: // Main training loop

11: **for** $i = 1, 2, \ldots$ **do**

12:   Sample mini-batch $b_{\text{src}} := \{(s, a, r, s', \text{timestep}, \text{mask}, d^m, d^w)\}$ with size $\frac{N}{2}$ from $\widehat{\mathcal{D}}_{\text{src}}$

13:   Sample mini-batch $b_{\text{tar}} := \{(s, a, r, s', \text{timesteps}, \text{masks})\}$ with size $\frac{N}{2}$ from $\widehat{\mathcal{D}}_{\text{tar}}$

14:   Normalize the deviations $d^w$ via Eq. (30) to obtain normalized deviations $\hat{d}^w$

15:   // Two-level data filtering

16:   Rank the MMD deviations of the sampled source domain data according to $d^m$ and admit the top $\xi\%$ of them

17:   Compute the OT weights for the remaining source domain data via $\exp(\eta^w \hat{d}^w)$

18:   Optimize the state-action value function $Q_\phi$ on $b_{\text{src}} \cup b_{\text{tar}}$ with multi-step TD targets via Eq. 8:

$$\mathcal{L}_Q(\phi; \mathcal{D}_{\text{tar}}) + \mathop{\mathbb{E}}_{\substack{\mathcal{T} \sim \mathcal{D}_{\text{src}} \\ \hat{a}_t \sim \pi_{\theta'}}} \left[ I_m(\tau^S) \sum_{i=t-K+1}^{t-1} \left\| \exp(\eta_w d^w_i)(\hat{Q}_i - Q_{\phi_i}(s_i, a_i)) \right\|^2 \right].$$

19:   Update the target network via $\phi' \leftarrow \eta_{\text{exp}}\phi + (1 - \eta_{\text{exp}})\phi'$

20:   // Policy adaptation

21:   Optimize the policy $\pi$ on $b_{\text{src}} \cup b_{\text{tar}}$ using the Q-guided loss function:

$$\mathcal{L}_\pi = \mathcal{L}^w_{\text{DT}} - \alpha \cdot \mathbb{E}_{\tau \in \mathcal{D}_{\text{tar}} \cup \mathcal{D}_{\text{src}}} \left[ \frac{I_m(\tau^S)}{K} \sum_{i=t-K+1}^{t} \exp(\eta_w d^w_i) Q_\phi(s, \pi(s)) \right].$$

22: **end for**

---

## F  ALGORITHM DETAILS OF DFDT

**Computing method of normalized scores.** Because raw returns are not directly comparable across environments, we follow D4RL (Fu et al., 2020) and report the Normalized Score (NS):

$$\text{NS} = \frac{\hat{J} - \hat{J}_{\text{rand}}}{\hat{J}_{\text{exp}} - \hat{J}_{\text{rand}}} \times 100, \tag{29}$$

where $\hat{J}$ is the empirical return of the learned policy, $\hat{J}_{\text{exp}}$ is the expert policy's empirical return, and $\hat{J}_{\text{rand}}$ is the empirical return of a random policy. By construction, $\text{NS} = 100$ corresponds to expert-level performance and $\text{NS} = 0$ corresponds to random performance. See Appendix C.1 of Lyu et al. (2025b) for dataset details about $\hat{J}_{\text{rand}}$ and $\hat{J}_{\text{exp}}$.

---

**Algorithm 2** DFDT Inference

---

**Require:** Trained DT policy $\pi_\phi$, trained command network $C_\omega$, sequence length $K$, (optional) normalization stats $(\mu_A, \sigma_A)$ from training, environment $\mathcal{M}_T$

1: // No critics or OT/MMD are needed at test time. We only use $C_\omega$ to produce command tokens and $\pi_\phi$ to act.
2: Initialise circular buffers for the last $K$ tokens:

$$\mathsf{S} \leftarrow [\,], \quad \mathsf{A} \leftarrow [\,], \quad \mathsf{C} \leftarrow [\,], \quad \mathsf{T} \leftarrow [\,], \quad \mathsf{M} \leftarrow [\,]$$

3: Reset environment; receive initial state $s_1$ and set $t \leftarrow 1$
4: **while** episode not terminal **do**
5:    // Compute command token from the current state
6:    $c_t^{\mathrm{raw}} \leftarrow C_\omega(s_t)$
7:    **if** training used standardized advantages (cf. Eq. ( 33 )) **then** $\quad c_t \leftarrow c_t^{\mathrm{raw}}$ **else** $\quad c_t \leftarrow \dfrac{c_t^{\mathrm{raw}} - \mu_A}{\sigma_A + \varepsilon}$ **end if**
8:    // Update rolling context (pad left with zeros and mask invalid tokens)
9:    Append $s_t$ to $\mathsf{S}$, $c_t$ to $\mathsf{C}$, $t$ to $\mathsf{T}$, and $1$ to $\mathsf{M}$; keep only the last $K$ entries of each
10:   Let $\mathsf{S}_{t-K+1:t}$, $\mathsf{C}_{t-K+1:t}$, $\mathsf{T}_{t-K+1:t}$, $\mathsf{M}_{t-K+1:t}$ be the length-$K$ sequences after left-padding with zeros;
11:   Define $\mathsf{A}_{t-K+1:t-1}$ as the last $K-1$ actions (left-padded with zeros); if $t = 1$ then $\mathsf{A}_{t-K+1:t-1}$ is all zeros and the first mask entries in $\mathsf{M}$ are $0$
12:   // Policy inference with command-conditioned tokens
13:   $a_t \leftarrow \pi_\phi\big(\mathsf{S}_{t-K+1:t}, \mathsf{A}_{t-K+1:t-1}, \mathsf{C}_{t-K+1:t}, \mathsf{T}_{t-K+1:t}, \mathsf{M}_{t-K+1:t}\big)$
14:   Execute $a_t$ in $\mathcal{M}_T$; observe $(r_t, s_{t+1})$
15:   Append $a_t$ to $\mathsf{A}$ (keep last $K-1$); set $t \leftarrow t + 1$ and $s_t \leftarrow s_t$
16: **end while**
17: **return** trajectory $\tau = \{(s_t, a_t, r_t)\}_{t=1}^T$

---

**Normalisation of OT-based deviations.** To make the OT-derived deviations $d_i^w$ numerically stable across tasks and batches, we apply a min–max normalisation that shifts the range to $[-1, 0]$:

$$\hat{d}_i^w := \frac{d_i^w - \max_{j \in \mathcal{D}_{\mathrm{src}}} d_j^w}{\max_{j \in \mathcal{D}_{\mathrm{src}}} d_j^w - \min_{j \in \mathcal{D}_{\mathrm{src}}} d_j^w}. \tag{30}$$

This mapping guarantees $\hat{d}_i^w \in [-1, 0]$, hence the exponential weights

$$w_i := \exp(\eta_w \hat{d}_i^w) \in \left[e^{-\eta_w}, 1\right] \tag{31}$$

are bounded, preventing gradient explosion while still down-weighting OT-distant (less feasible) source fragments. Practically, Eq. (30) makes weighting scale-free across domains and robust to outliers in $d^w$. We use $w_i$ both in critic fitting and in weighted DT losses on the source batch (see blue terms in Alg. 1).

**Command network $C_\omega$ trained via expectile regression.** The command network $C_\omega$ outputs an *advantage-consistent command token* that replaces RTG during inference. We first form per-token advantages from the auxiliary value estimators,

$$A_i := Q_\psi(s_i, a_i) - V_\varphi(s_i), \tag{32}$$

and optionally standardize them within $\mathcal{D}_{\mathrm{src}}$ to improve numerical stability:

$$\tilde{A}_i := \frac{A_i - \mu_A}{\sigma_A + \varepsilon}, \tag{33}$$

where $(\mu_A, \sigma_A)$ are the mean and standard deviation of $\{A_i\}$. We then train $C_\omega$ to predict a high-expectile summary of the state-conditional advantage distribution using the asymmetric least-squares (ALS) loss:

$$\mathcal{L}_C(\omega) = \mathbb{E}_{(s_i, a_i) \sim \mathcal{D}_{\mathrm{tar}} \cup \mathcal{D}_{\mathrm{src}}} \left[ \left| \zeta - \mathbf{1}\{\tilde{A}_i - C_\omega(s_i) < 0\} \right| \left( \tilde{A}_i - C_\omega(s_i) \right)^2 \cdot \left( I_m(\tau^S) w_i \right) \right], \tag{34}$$

where $\zeta \in (0.5, 1)$ (e.g., $0.7 \sim 0.9$) controls the degree of optimism. The ALS penalty $\rho_\zeta(u) = |\zeta - \mathbf{1}\{u < 0\}|u^2$ penalizes over-predicting low-advantage samples more than under-predicting high-advantage ones, so that $C_\omega$ captures a smooth, optimistic summary of advantages. The factor $I_m(\tau^S) w_i$ further ensures that $C_\omega$ is mainly trained on MMD-filtered and OT-weighted tokens that are compatible with the target dynamics.

**Usage of $C_\omega$ at inference.** At test time, we compute a scalar command $c_t = C_\omega(s_t)$ from the current state and feed it as the conditioning token to the DT in place of RTG. Because $c_t$ is derived from standardized advantages rather than raw returns, it offers a reward- and horizon-agnostic guidance signal that is shared across domains. This stabilizes token-level conditioning under cross-domain shifts and reduces stitching artefacts caused by inconsistent reward scales or episode lengths.

Table 3: Default hyperparameter setup for DFDT. For exact hyperparameter setups for each experiment, please refer to the code base.

| Hyperparameter | Value |
|---|---|
| Number of layers | 4 |
| Number of attention heads | 4 |
| Embedding dimension | 256 |
| Context length $K$ | $\{5, 10, 20\}$ |
| Dropout | 0.1 |
| Learning rate | $3 \times 10^{-4}$ |
| Optimizer | Adam (Kingma & Ba, 2015) |
| Discount factor | 0.99 |
| Nonlinearity | ReLU |
| Target update rate | $5 \times 10^{-3}$ |
| Pretrained Q network hidden size | $(256, 256, 256)$ |
| Pretrained V network hidden size | $(256, 256, 256)$ |
| Command network hidden size | $(256, 256, 256)$ |
| Number of sampled latent variables $M$ | 10 |
| Standard deviation of Gaussian distribution | $\sqrt{0.1}$ |
| OT Cost function | cosine |
| Data filtering ratio $\xi\%$ | 50% |
| Policy regularization coefficient $\eta_{\text{reg}}$ | $\{0.3, 0.4, 0.5, 0.6\}$ |
| Source domain Batch size | 64 |
| Target domain Batch size | 128 |

# G    ENVIRONMENT DETAILS AND SETUP

## G.1    ENVIRONMENT SPECIFICATIONS

Following recent work in offline reinforcement learning (Fu et al., 2020) and cross-domain policy adaptation (Lyu et al., 2025b), we evaluate DFDT on four continuous control tasks from the MuJoCo simulator (Todorov et al., 2012): `HalfCheetah-v2`, `Hopper-v2`, `Walker2d-v2`, and `Ant-v2`. All environments use the MuJoCo v2 physics engine via OpenAI Gym (Brockman et al., 2016).

We use the standard D4RL configuration with a maximum episode length of 1000 timesteps for all tasks. Episodes terminate early only if the agent enters a failure state (e.g., the Hopper falls over). We adopt the original D4RL reward specifications without modification, including implicit survival bonuses and center-of-mass velocity components where applicable.

**Transformer context length.**    Due to varying trajectory structure across tasks, we use set task-specific context windows. For morphology and kinematic shifts, we used $K = 20$ for the HalfCheetah task and $K = 5$ for the Hopper, Walker2D and Ant tasks. For gravity shifts, we used $K = 10$ for the HalfCheetah task, $K = 20$ for the Hopper and Walker2D, and $K = 5$ for the Ant task. These results help balance model capacity and empirical performance in dynamic shifts.

## G.2 EVALUATION PROTOCOL

**Normalized score.** For each trained policy, we report the normalized score (NS) averaged over 10 evaluation episodes using deterministic actions (mean of the policy distribution). The normalized score is computed as:

$$\text{NS} = \frac{\hat{J} - \hat{J}_{\text{rand}}}{\hat{J}_{\text{exp}} - \hat{J}_{\text{rand}}} \times 100, \tag{35}$$

where $\hat{J}$ is the empirical return of the learned policy, $\hat{J}_{\text{exp}}$ is the expert policy's return in the *target domain*, and $\hat{J}_{\text{rand}}$ is the random policy's return in the target domain. Reference values for $\hat{J}_{\text{exp}}$ and $\hat{J}_{\text{rand}}$ under each domain shift are provided in Appendix C.1 of Lyu et al. (2025b).

**Random seeds.** For main experiments (Tables 1, 2, 4), we evaluate policies using the final checkpoint from each of 5 random seeds (1, 2, 3, 4, 5) and report the mean ± standard deviation across seeds. Ablation studies use different evaluation protocols as specified in individual table captions.

**Computational resources.** Each experiment is trained on a single NVIDIA RTX 4090 GPU (24 GB). The total training time per task is typically 6–7 hours for 100,000 gradient steps (see Appendix J for detailed training time analyses).

**Datasets.** Our source-domain datasets are drawn from the D4RL benchmark (Fu et al., 2020). Target-domain datasets are constructed under three types of dynamics shifts: **gravity shifts**, **kinematic shifts**, and **morphology shifts**. The exact environment modifications (XML parameter changes, joint range restrictions, body geometry alterations) follow Appendix C.2–C.4 of Lyu et al. (2025b) exactly. We refer readers to their work for detailed XML code and implementation specifics.

**State normalization.** For all DFDT experiments, we apply per-dimension normalization to state observations:

$$\hat{s}_i = \frac{s_i - \mu_i}{\sigma_i + \epsilon}, \tag{36}$$

where $\mu_i$ and $\sigma_i$ are the mean and standard deviation of dimension $i$ computed over all states in the target training dataset, and $\epsilon = 10^{-8}$ prevents division by zero. This normalization is applied identically during training and evaluation.

We normalize states but *not* actions or rewards. The normalization statistics are computed once before training and remain fixed throughout all experiments for a given source-target pair.

**Two-stage data filtering.** We filter source data in two sequential stages to balance data utilization and cross-domain compatibility.

**Cost computations before the main training loop.** Before training begins, we pre-compute the following quantities:

- Maximum Mean Discrepancy (MMD) between each source episode $((s_t^S, a_t^S, s_{t+1}^S, r_t^S), (s_{t+1}^S, a_{t+1}^S, s_{t+2}^S, r_{t+1}^S), \ldots, (s_{t+K}^S, a_{t+K}^S, s_{t+K+1}^S, r_{t+K+1}^S))$ and each episode in the target dataset $((s_t^T, a_t^T, s_{t+1}^T, r_t^T), (s_{t+1}^T, a_{t+1}^T, s_{t+2}^T, r_{t+1}^T), \ldots, (s_{t+K}^T, a_{t+K}^T, s_{t+K+1}^T, r_{t+K+1}^T))$.

- Optimal transport (OT) distances from each source transition $(s_t^S, a_t^S, r_t^S, s_{t+1}^S)$ to each target transition $(s_t^T, a_t^T, r_t^T, s_{t+1}^T)$.

Note that while Section 3.1.1 presents the framework using state features $f_\phi(s)$ for clarity, we encode full transitions $f_\phi(s, a, s', r)$ in practice. Table 8 demonstrates this choice performs comparably to state-only features, validating that our implementation preserves the intended function of the MMD component.

**Stage 1 (MMD filtering for fragment selection).** We first rank all source transitions by their MMD distance and retain only the $\xi_{\text{MMD}}\% = 50\%$ with the lowest MMD, yielding a pre-filtered source dataset $\mathcal{D}_{\text{src}}^{\text{MMD}}$.

**Stage 2 (OT filtering for action feasibility).** During training, when constructing each mini-batch, we sample 128 source transitions from $\mathcal{D}_{\text{src}}^{\text{MMD}}$ and use their pre-computed OT distances to retain only the top $\xi_{\text{OT}}\% = 50\%$ with the lowest OT distance. These filtered source transitions are then combined with 128 target transitions, giving a final batch size of 192.

**Hyperparameter overview.** Table 3 summarises the hyperparameters of a compact Transformer backbone for DFDT (multi-head attention with moderate depth, width, and context length), trained with standard optimisation and stabilisation choices (Adam, dropout, ReLU, soft target updates, and a fixed discount factor). Method-specific settings for DFDT include pretrained critics, a command network, and an OT-based filtering module (cosine cost) coupled with a fixed fragment-filtering ratio and a source–target mixing coefficient. We adopt asymmetric batch sizes to emphasise target-domain learning while still leveraging filtered source fragments. The defaults were chosen from small grids over context length and mixing strength and were found to be robust across seeds and tasks.

Table 4: **Performance comparison of cross-domain offline RL algorithms given gravity shifts.** The meanings of each abbreviation are the same as those listed in Table 1. We report *normalized* target-domain performance (*mean ± std.*) over **five** seeds; We **bold** and highlight the best cell.

| Source | Target | IQL | DARA | IGDF | OTDF | DT | QT | DADT | DFDT |
|---|---|---|---|---|---|---|---|---|---|
| half-m | medium | 39.6 | **41.2** | 36.6 | 40.7 | 28.4 | 40.2 | 36.6 | 7.3±4.3 |
| half-m | medium-expert | 39.6 | 40.7 | 38.7 | 28.6 | 45.1 | **62.1** | 34.7 | 7.8±2.4 |
| half-m | expert | 42.4 | 39.8 | 39.6 | 36.1 | 41.8 | **49.1** | 45.7 | 13.8±11.7 |
| half-m-r | medium | 20.1 | 17.6 | 14.4 | 21.5 | 18.3 | **51.6** | 25.3 | 5.9±2.5 |
| half-m-r | medium-expert | 17.2 | 20.2 | 10.0 | 14.7 | 17.2 | 2.1 | **27.1** | 5.7±2.4 |
| half-m-r | expert | 20.7 | 22.4 | 15.3 | 11.4 | 7.8 | 2.5 | **23.6** | 17.9±10.0 |
| half-m-e | medium | 38.6 | 37.8 | 37.7 | 39.5 | 35.1 | **69.3** | 44.0 | 5.6±2.6 |
| half-m-e | medium-expert | 39.6 | 39.4 | 40.7 | 32.4 | 38.2 | **67.0** | 32.0 | 6.0±2.9 |
| half-m-e | expert | 43.4 | 45.3 | 41.1 | 26.5 | 40.7 | **68.5** | 37.8 | 21.9±8.3 |
| hopp-m | medium | 11.2 | 17.3 | 15.3 | 32.4 | 19.7 | 16.1 | 12.8 | **82.4**±7.5 |
| hopp-m | medium-expert | 14.7 | 15.4 | 15.1 | 24.2 | 11.6 | 12.8 | 11.6 | **56.7**±23.3 |
| hopp-m | expert | 12.5 | 19.3 | 14.8 | **33.7** | 11.0 | 12.3 | 12.7 | 22.7±11.2 |
| hopp-m-r | medium | 13.9 | 10.7 | 15.3 | 31.1 | 14.2 | 19.9 | 22.6 | **58.8**±27.5 |
| hopp-m-r | medium-expert | 13.3 | 12.5 | 15.4 | 24.2 | 13.7 | 22.3 | 16.6 | **66.4**±17.7 |
| hopp-m-r | expert | 11.0 | 14.3 | 16.1 | 31.0 | 19.6 | 18.7 | 21.5 | **42.4**±16.6 |
| hopp-m-e | medium | 19.1 | 18.5 | 22.3 | 26.4 | 13.0 | 14.3 | 11.6 | **63.6**±18.0 |
| hopp-m-e | medium-expert | 16.8 | 16.0 | 16.6 | 28.3 | 13.6 | 14.4 | 11.7 | **39.2**±27.8 |
| hopp-m-e | expert | 20.9 | 23.9 | 26.0 | 44.9 | 13.1 | 14.0 | 13.2 | **75.4**±19.0 |
| walk-m | medium | 28.1 | 28.4 | 22.1 | 36.6 | 36.2 | 29.5 | 37.4 | **43.1**±7.2 |
| walk-m | medium-expert | 35.7 | 30.7 | 35.4 | 44.8 | 38.2 | **45.2** | 29.1 | 21.5±4.5 |
| walk-m | expert | 37.3 | 36.0 | 36.2 | 44.0 | 46.4 | 44.0 | **54.0** | 22.6±5.7 |
| walk-m-r | medium | 14.6 | 14.1 | 11.6 | 32.7 | 28.6 | 18.9 | 24.8 | **44.1**±2.9 |
| walk-m-r | medium-expert | 15.3 | 15.9 | 13.9 | **31.6** | 26.9 | 20.0 | 29.8 | 22.7±7.0 |
| walk-m-r | expert | 15.8 | 15.7 | 15.2 | **31.3** | 28.0 | 28.6 | 20.1 | 26.7±11.8 |
| walk-m-e | medium | 39.9 | 41.6 | 33.8 | 30.2 | 42.5 | **56.7** | 45.5 | 41.4±3.2 |
| walk-m-e | medium-expert | 49.1 | 45.8 | 44.7 | 53.3 | 39.4 | **55.8** | 30.6 | 23.6±5.1 |
| walk-m-e | expert | 40.4 | 56.4 | 45.3 | **61.1** | 39.6 | 47.4 | 34.5 | 23.6±8.9 |
| ant-m | medium | 10.2 | 9.4 | 11.3 | 45.1 | 22.0 | 15.3 | 12.4 | **61.0**±8.7 |
| ant-m | medium-expert | 9.4 | 10.0 | 9.4 | 33.9 | 17.7 | 14.1 | 14.0 | **52.8**±15.7 |
| ant-m | expert | 10.2 | 9.8 | 9.7 | 33.2 | 18.9 | 15.7 | 13.7 | **58.3**±5.8 |
| ant-m-r | medium | 18.9 | 21.7 | 19.6 | 29.6 | 18.8 | 13.9 | 21.4 | **66.9**±8.5 |
| ant-m-r | medium-expert | 19.1 | 18.3 | 20.3 | 25.4 | 13.9 | 13.6 | 18.5 | **44.9**±5.5 |
| ant-m-r | expert | 18.5 | 20.0 | 18.8 | 24.5 | 14.6 | 10.6 | 17.7 | **38.8**±11.1 |
| ant-m-e | medium | 9.8 | 8.1 | 8.9 | 18.6 | 11.3 | 11.6 | 20.6 | **68.9**±6.4 |
| ant-m-e | medium-expert | 9.0 | 6.4 | 7.2 | 34.0 | 18.0 | 12.2 | 15.2 | **45.7**±18.0 |
| ant-m-e | expert | 9.1 | 10.4 | 9.2 | 23.2 | 11.6 | 10.0 | 15.3 | **41.2**±11.9 |
| | Total Score | 825.0 | 851.0 | 803.6 | 1160.7 | 874.7 | 1020.3 | 895.7 | **1347.3** |

# H  WIDER MAIN EXPERIMENTAL RESULTS UNDER GRAVITY SHIFTS

We further report comprehensive results for gravity shifts in Table 4. DFDT attains the best mean performance on **19** out of **36** tasks and achieves the highest total normalized score of **1347.3**, exceeding IQL by **63.3**% (1347.3 vs. 825.0), the second–best approach OTDF by **16.1**% (1347.3 vs. 1160.7), and the strong sequence baseline QT by **32.0**% (1347.3 vs. 1020.3). Breaking down by environment family, DFDT dominates *hopper* (wins **8** out of **9**) and *ant* (wins **9** out of **9**), remains competitive on the *walker2d* task (wins **2** out of **9**), while *halfcheetah* is largely led by QT. Notably, DFDT delivers large margins in challenging settings such as `hopp-grav-me2e` ( **75.4** ± 19.0 ) and `ant-grav-m2m` ( **61.0** ± 8.7 ), where we define grav = gravity, kinematic = kin, morph = morphology, reflecting robust cross-dynamics stitching. Overall, these results corroborate DFDT's offline policy adaptation strength under gravity shifts, complementing its competitiveness on the remaining tasks.

# I  ABLATION EXPERIMENTS

## I.1  ABLATION SETUP

For the ablation studies, we start from the default DFDT configuration and vary one component or hyperparameter at a time, keeping all other settings identical to the main experiments. To control computational cost, we evaluate on a small set of representative source–target pairs that cover all three shift types (kinematic, gravity, morphology); the exact tasks are listed in the corresponding tables.

Concretely, we (i) compare full DFDT (MMD+OT) against MMD-only and OT-only variants and sweep the MMD keep-ratio $\xi$ used by the state-structure gate; (ii) replace advantage-conditioned tokens with RTG tokens; (iii) vary the Q-regularization coefficient $\alpha$; (iv) sweep the expectile coefficient $\zeta$ used in the command network; and (v) modify the OT design by including or excluding the 1-dim reward channel $r$ and, in a preliminary study, by changing the OT cost function (cosine, Euclidean, squared Euclidean).

Table 5: Ablation study on data filtering.

| Source | Target | Shift | DFDT | OT-only | MMD-only |
|---|---|---|---|---|---|
| ant-m | medium | gravity | **63.8** ± 1.4 | 59.6 ± 6.1 | 62.6 ± 2.4 |
| hopp-m | medium | kinematic | **66.5** ± 0.9 | 17.3 ± 8.8 | 64.2 ± 1.5 |
| hopp-m-r | medium | morph | **56.7** ± 0.0 | 50.0 ± 1.1 | 49.0 ± 2.7 |
| walk-m | medium | morph-expert | **41.1** ± 5.3 | 31.7 ± 1.1 | 40.4 ± 0.0 |

## I.2  ABLATION ON FRAGMENT FILTERING

We next study how the two-stage data filtering affects performance by comparing DFDT (MMD+OT) with OT-only and MMD-only variants. All results are averaged over two random seeds, with each seed's performance computed as the mean over the last three checkpoints; entries with zero standard deviation correspond to a single available seed.

Table 5 compares DFDT with OT-only and MMD-only filtering on four representative source–target pairs. Across all four settings, DFDT achieves the best return, showing that combining state-distribution matching (MMD) with action-feasibility reweighting (OT) yields the most robust data selection. On the three more challenging shifts (`ant-m`→`medium` (gravity), `hopp-m`→`medium` (kinematic), `hopp-m-r`→`medium` (morphology)), DFDT improves over OT-only by roughly +4.2, +49.2, and +6.7 points, respectively, and also provides consistent gains over MMD-only (+1.2, +2.3, and +7.7 points). This indicates that relying solely on OT can be unstable under large state-structure shifts, e.g., `hopp-m`→`medium` (kinematic), where the lack of an explicit state manifold gate can lead to catastrophic failures, while augmenting OT with MMD-based state alignment effectively prevents such breakdowns. MMD-only is clearly stronger than OT-only and often close to DFDT, highlighting that state-structure alignment is already a key ingredient for cross-dynamics transfer. In the milder `walk-m`→`medium` (morphology, expert target) setting, MMD-only nearly

matches DFDT (40.4 vs. 41.1) and both are noticeably better than OT-only (31.7), suggesting that when the mismatch is moderate, MMD-based state filtering dominates, whereas under severe dynamic shifts the full MMD+OT scheme is necessary to recover stitchable and high-performing trajectories.

## I.3 ABLATION ON ADVANTAGE-CONDITIONED TOKENS

Table 6 shows that replacing RTG tokens with advantage-conditioned tokens consistently improves DFDT under kinematic shifts. All results are averaged over two random seeds, with each seed's performance computed as the mean over the last three checkpoints. Across all three source–target pairs, the advantage-conditioned variant outperforms the RTG-based one by +7.0 (`ant-m`→`medium` (kinematic), 59.2 vs. 52.2), +1.9 (`hopp-m`→`medium` (kinematic), 66.5 vs. 64.6), and a substantial +28.8 points on the more demanding `hopp-m`→`medium-expert` (kinematic) task (78.5 vs. 49.7). These gains indicate that encoding *relative* value information (advantages) provides a sharper, scale-robust conditioning signal than absolute RTG levels, especially when mixing demonstrations of heterogeneous quality across domains. In particular, when the target dataset is near-expert, RTG tokens can saturate and lose resolution, whereas advantage-conditioned tokens still highlight locally high-improvement decisions along a trajectory. Overall, this ablation confirms that advantage-based conditioning is better suited for cross-domain token selection and leads to more stable and effective adaptation than RTG-based conditioning.

Table 6: Ablation study on advantage-conditioned tokens.

| Source | Target | Shift | Advantage | RTG |
|--------|--------|-------|-----------|-----|
| ant-m | medium | kinematic | **59.2** $\pm$ 2.0 | 52.2 $\pm$ 2.7 |
| hopp-m | medium | kinematic | **66.5** $\pm$ 0.9 | 64.6 $\pm$ 2.3 |
| hopp-m | medium-expert | kinematic | **78.5** $\pm$ 2.6 | 49.7 $\pm$ 24.8 |

## I.4 ABLATION ON Q-REGULARIZATION COEFFICIENT

We ablate the Q-regularization coefficient $\alpha$ on a range of cross-dynamics tasks (Table 7). Performance exhibits clear sensitivity to this hyperparameter, with notable variation across tasks and shift types. On several challenging tasks, non-zero regularization provides substantial benefits: for instance, on `ant-m`→`medium` (kinematic), performance improves from 50.5 $\pm$ 4.5 at $\alpha = 0.0$ to 59.2 $\pm$ 2.0 at $\alpha = 3.5$, and on `walk-m-e`→`medium` (kinematic), scores rise from 46.2 $\pm$ 12.5 to 59.6 $\pm$ 7.8 at $\alpha = 5.0$, indicating that penalizing $Q$-jumps at fragment junctions stabilizes token stitching under challenging dynamics shifts.

Table 7: Ablation study on Q-regularization coefficient $\alpha$ for DFDT. Results show the mean and standard deviation over the last 3 checkpoints from a single seed.

| Source | Target | Shift | $\alpha = 0.0$ | $\alpha = 1.0$ | $\alpha = 3.5$ | $\alpha = 5.0$ |
|--------|--------|-------|----------------|----------------|----------------|----------------|
| half-m | medium | morph | 43.9 $\pm$ 0.4 | **44.2** $\pm$ 0.1 | 43.3 $\pm$ 0.2 | 43.9 $\pm$ 0.1 |
| half-m-e | medium | morph | **44.4** $\pm$ 0.1 | 44.2 $\pm$ 0.4 | 44.2 $\pm$ 0.6 | 44.1 $\pm$ 0.2 |
| hopp-m-e | medium | morph | 25.1 $\pm$ 1.0 | 27.1 $\pm$ 2.2 | 56.5 $\pm$ 25.7 | **93.5** $\pm$ 4.7 |
| hopp-m-e | medium | kinematic | 62.3 $\pm$ 2.4 | **65.8** $\pm$ 1.9 | 63.1 $\pm$ 2.2 | 63.4 $\pm$ 0.9 |
| hopp-m | medium | gravity | **82.4** $\pm$ 7.5 | 59.8 $\pm$ 22.3 | 60 $\pm$ 22.3 | 42.3 $\pm$ 7.3 |
| hopp-m-e | medium | gravity | 61.7 $\pm$ 32.3 | 62.3 $\pm$ 17.0 | 61.2 $\pm$ 27.3 | **63.6** $\pm$ 18.0 |
| walk-m-e | medium | morph | 42.3 $\pm$ 8.3 | 41.2 $\pm$ 13.1 | **51.4** $\pm$ 9.2 | 45.6 $\pm$ 3.0 |
| walk-m-e | medium | kinematic | 46.2 $\pm$ 12.5 | 50.4 $\pm$ 14.3 | 47.4 $\pm$ 21.9 | **59.6** $\pm$ 7.8 |
| ant-m | medium | kinematic | 50.5 $\pm$ 4.5 | 56.6 $\pm$ 3.3 | **59.2** $\pm$ 2.0 | 53.2 $\pm$ 7.7 |
| ant-m-e | medium | gravity | 62.4 $\pm$ 1.1 | 65.1 $\pm$ 5.1 | 63.8 $\pm$ 9.5 | **68.9** $\pm$ 6.4 |

However, the optimal value is highly task-dependent and no single setting dominates across all environments. For example, `hopp-m`→`medium` (gravity) achieves its best performance with no regularization (82.4 $\pm$ 7.5 at $\alpha = 0.0$), while `hopp-m-e`→`medium` (morphology) peaks at strong regularization (93.5 $\pm$ 4.7 at $\alpha = 5.0$). Other tasks such as `half-m`→`medium` (morphology)

show minimal variation across all settings (43.3–44.2), suggesting that their value landscapes are intrinsically smoother and less sensitive to junction regularization.

Among the tested values, moderate regularization ($\alpha = 3.5$) achieves the best or near-best performance on several key tasks, including `ant-m→medium` (kinematic, $59.2 \pm 2.0$) and `walk-m-e→medium` (morphology, $51.4 \pm 9.2$), while avoiding the performance degradation observed with extreme choices on other environments. Based on this trade-off between stability and flexibility across diverse shift types, we adopt $\alpha = 3.5$ as our default configuration for the main experiments, recognizing that task-specific tuning could yield further gains but at the cost of generality.

Table 8: Ablation study on the data used for MMD calculations. For each seed, we compute the mean over the last 3 checkpoints; reported values are mean $\pm$ std across the 3 per-seed means.

| Source | Target | Shift | State | Transition | Transition-reward |
|---|---|---|---|---|---|
| hopp-m-r | medium | morph | $52.5 \pm 23.3$ | $66.5 \pm 13.4$ | **$68.2 \pm 12.8$** |
| half-m | medium | morph | $42.2 \pm 0.9$ | **$43.0 \pm 0.6$** | $41.8 \pm 1.1$ |
| ant-m-e | medium | gravity | $62.6 \pm 7.9$ | $62.8 \pm 3.5$ | **$64.3 \pm 3.3$** |
| walk-m-e | medium | kinematic | $58.2 \pm 6.3$ | **$58.5 \pm 7.7$** | $55.9 \pm 5.1$ |

## I.5 ABLATION ON MMD FEATURE SELECTION

We ablate which features to feed into the MMD-based state-structure filter by comparing three variants (Table 8): *State* (only $(s_n^S, \ldots, s_{n+K}^S)$), *Transition* (adding actions $(s_n^S, a_n^S, \ldots, s_{n+K}^S, a_{n+K}^S)$), and *Transition-reward* (further adding rewards $(s_n^S, a_n^S, r_n^S, \ldots, s_{n+K}^S, a_{n+K}^S, r_{n+K}^S)$).

Overall, DFDT is reasonably robust to this choice. On `half-m→medium` (morphology), all three settings are tightly clustered around 42–43 ($42.2 \pm 0.9$, $43.0 \pm 0.6$, $41.8 \pm 1.1$), and the gravity/kinematic cases `ant-m-e→medium` and `walk-m-e→medium` also differ by only a few points with overlapping variances ($62.6 \pm 7.9$, $62.8 \pm 3.5$, $64.3 \pm 3.3$ and $58.2 \pm 6.3$, $58.5 \pm 7.7$, $55.9 \pm 5.1$). The main gains appear on the harder morphology shift `hopp-m-r→medium`, where enriching the features from state-only to transition-reward lifts the mean from 52.5 to 68.2 and clearly reduces variance (23.3 vs. 12.8).

These results suggest that state-only MMD already provides a solid baseline, but incorporating actions and rewards can substantially stabilize filtering on challenging morphology shifts. We therefore use *transition-reward* MMD as the default in our implementation: it delivers the best or near-best mean performance across shift types while keeping the method only moderately sensitive to the exact feature choice, leaving room for practitioners to adapt the feature set to computational or domain-specific constraints.

Table 9: Ablation study on the expectile coefficient for DFDT. Results show the mean and standard deviation over the last 3 checkpoints from a single seed.

| Source | Target | Shift | $\zeta = 0.5$ | $\zeta = 0.55$ | $\zeta = 0.7$ | $\zeta = 0.98$ |
|---|---|---|---|---|---|---|
| half-m-e | medium | kinematic | $41.5 \pm 0.2$ | $41.5 \pm 0.3$ | **$41.7 \pm 0.4$** | $41.5 \pm 0.3$ |
| ant-m | medium | gravity | **$61.8 \pm 4.3$** | $58 \pm 2.9$ | $60.5 \pm 16.3$ | $58.5 \pm 3.8$ |
| walk-m | medium-expert | morph | $31.1 \pm 16.1$ | $32.6 \pm 3.4$ | $39.7 \pm 6.6$ | **$41.1 \pm 5.3$** |
| hopp-m-e | medium | kinematic | $61.9 \pm 3.0$ | $61.4 \pm 2.7$ | $61.0 \pm 3.3$ | **$62.2 \pm 2.7$** |

## I.6 ABLATION ON EXPECTILE COEFFICIENT

We study the expectile coefficient $\zeta$ used in the command network's expectile regression (Table 9). Across most tasks, DFDT is fairly insensitive to this choice. For example, on `half-m-e→medium` (kinematic) all settings yield almost identical performance (41.5–41.7), and on `hopp-m-e→medium` (kinematic) scores remain in a narrow band around 61–62, with $\zeta = 0.98$ only slightly ahead. For `ant-m→medium` (gravity), all four values stay within a few points of the best setting $61.8 \pm 4.3$, showing no sharply tuned optimum.

The main sensitivity appears on `walk-m→medium-expert` (morphology), where more optimistic expectiles substantially help: the mean return increases from $31.1 \pm 16.1$ at $\zeta = 0.5$ to $41.1 \pm 5.3$ at $\zeta = 0.98$. Overall, these trends indicate that DFDT remains robust over a broad range of $\zeta$ while moderately benefiting from higher, more optimistic values on harder morphology shifts. We therefore adopt $\zeta = 0.98$ as the default, as it is best or competitive across all tested tasks while encouraging sufficiently optimistic value-based conditioning.

Table 10: Ablation study on the 1-dim $r$ channel of OT computation. For each seed, we compute the mean over the last 3 checkpoints; reported values are mean $\pm$ std across the 3 per-seed means.

| Source | Target | Shift | OT w the $r$ channel | OT w/o the $r$ channel |
|---|---|---|---|---|
| ant-m | medium | kinematic | $53.1 \pm 6.4$ | $\mathbf{56.9} \pm 3.1$ |
| hopp-m | medium | kinematic | $59.3 \pm 11.4$ | $\mathbf{63.7} \pm 3.6$ |
| hopp-m | medium | morph | $\mathbf{57.8} \pm 11.9$ | $53.0 \pm 13.4$ |
| half-m-r | medium-expert | morph | $43.8 \pm 0.8$ | $\mathbf{44.6} \pm 0.4$ |
| walk-m-r | medium | gravity | $40.7 \pm 4.1$ | $\mathbf{44.3} \pm 0.8$ |

## I.7 ABLATION OF THE 1-DIM REWARD CHANNEL IN THE OT COMPUTATION

We ablate the effect of including the 1-dim reward channel $r$ in the OT cost (Table 10). Across the five tested configurations, *excluding* $r$ improves performance on four tasks and remains competitive on the fifth. For example, on `ant-m→medium` and `hopp-m→medium` (both kinematic), dropping $r$ yields higher returns with notably lower variance ($56.9 \pm 3.1$ vs. $53.1 \pm 6.4$ and $63.7 \pm 3.6$ vs. $59.3 \pm 11.4$), and similar gains appear on the morphology and gravity cases `half-m-r→medium-expert` and `walk-m-r→medium`. Only `hopp-m→medium` (morphology) sees a modest benefit from including $r$, and both variants exhibit high variance there.

These results suggest that the structural information in $(s, a, s')$ is usually sufficient for effective OT-based alignment, while adding reward can inject extra noise, especially under kinematic shifts where action feasibility is the main concern. In our main experiments we retain $(s, a, s', r)$ for consistency with the original design, and leave more nuanced ways of exploiting reward information (e.g., task-dependent weighting or learned costs) to future work.

Table 11: Ablation study on cost functions used in OT. Results show the mean and standard deviation over the last 3 checkpoints from a single seed.

| Source | Target | Shift | Cosine | Euclidean | Squared Euclidean |
|---|---|---|---|---|---|
| ant-m | medium | gravity | $\mathbf{60.8} \pm 13.6$ | $59.6 \pm 9.1$ | $\mathbf{60.8} \pm 9.0$ |
| walk-m-r | medium | gravity | $36.5 \pm 5.3$ | $\mathbf{44.2} \pm 5.8$ | $43.2 \pm 4.5$ |
| hopp-m | medium | kinematic | $\mathbf{65.0} \pm 0.9$ | $60.8 \pm 5.5$ | $64.0 \pm 2.2$ |
| half-m | expert | kinematic | $\mathbf{30.7} \pm 9.8$ | $16.9 \pm 12.9$ | $15.6 \pm 4.9$ |
| half-m-e | medium | morph | $\mathbf{43.6} \pm 0.3$ | $42.9 \pm 1.1$ | $43.3 \pm 1.1$ |

## I.8 ABLATION ON OT COST FUNCTIONS

We ablate the cost function used in the optimal transport (OT) computation (Table 11), comparing cosine cost (our default, following OTDF (Lyu et al., 2025a)) with Euclidean and squared Euclidean variants. The results show moderate, task-dependent sensitivity to this choice. For gravity shifts, `ant-m→medium` is largely indifferent between cosine and squared Euclidean ($60.8 \pm 13.6$ vs. $60.8 \pm 9.0$), while `walk-m-r→medium` clearly prefers $\ell_2$-based costs (up to $44.2 \pm 5.8$) over cosine ($36.5 \pm 5.3$). For kinematic shifts, `hopp-m→medium` strongly favors cosine ($65.0 \pm 0.9$) over Euclidean variants, and `half-m→expert` also sees cosine dominate ($30.7 \pm 9.8$ vs. $16.9 \pm 12.9$ and $15.6 \pm 4.9$). On `half-m-e→medium` (morphology), all three costs are nearly tied around 43.

Overall, performance gaps are usually modest but clearly task- and shift-dependent, indicating that the learned embedding is fairly robust to cost formulation while still exhibiting geometry-specific preferences. We retain cosine cost as the default to stay consistent with OTDF and because it offers competitive or best performance on most tested tasks, while future work could explore adaptive or learned cost metrics tailored to specific shift types.

## J  TRAINING TIME ANALYSES

We report wall-clock time statistics for DFDT's training pipeline on two D4RL source datasets: `hop-grav-m2m` and `hc-morph-m2m`. Table 12 summarizes key phases:

- **Filtering phase**: Precomputes MMD/OT distances over source data (97.8-98.0 seconds per tested dataset).
- **Training phase**: Includes command network optimization (2,314-2,536 seconds) and VAE policy training (585-827 seconds), with total runtime 6-7 hours for 100k gradient updates.

Notably, the MMD and OT filtering step accounts for $\leq 0.5\%$ of total training time (98s vs. 24,000s), making it a negligible overhead compared to the full training process. This cost is justified by DFDT's ability to maintain token-level continuity across domain shifts, which standard DT baselines fail to achieve without such precomputation.

Table 12: Training time analyses for DFDT on D4RL datasets

| Dataset | Filtering Peak VRAM | Filtering Time | Training Peak VRAM | Command Network Time | VAE Policy Time | Total Training Time |
|---|---|---|---|---|---|---|
| hop-grav-m2m | 3638 MB | 97.8s | 2108 MB | 2,314.4s | 585.6s | 24,086.1s |
| hc-morph-m2m | 10398 MB | 98.0s | 4410 MB | 2,536.8s | 827.6s | 24,122.03s |

## K  LLM USAGE

The authors acknowledge that large language models (LLMs) were used in the following scenarios:

- **Ablation design and diagnostics.** ChatGPT (OpenAI, 2025) was consulted for ideas on diagnostic metrics, and suggested using action jumps, $Q$-value jumps, and TD residuals to visualize the stable, value-consistent sequence semantics of DFDT. The final choice of metrics and all implementation details were designed, implemented, and validated by the authors.
- **Code assistance.** Various models accessed through GitHub Copilot (GitHub, 2025) were used to assist in writing sections of the code base, including documentation, experimental utilities, and visualization scripts (e.g., the script for producing Figure 2). Copilot's autocomplete was also used for general programming assistance. All generated code was reviewed and, when necessary, modified by the authors.
- **Writing and formatting.** Templates for tables and algorithms were generated using ChatGPT based on screenshots of previous work such as figures from Lyu et al. (2025a). The contents of all tables and algorithms (including numerical values) were entered or verified by the authors. ChatGPT was further used to polish the language of the manuscript, while all technical claims, proofs, and conclusions were written or carefully checked by the authors.

