# OpenReview forum: "Data Fusion–Enhanced Decision Transformer for Stable Cross-Domain Generalization"
_ICLR.cc/2026/Conference — Submitted to ICLR 2026_

### Official Review · Reviewer_qmFQ · 2025-10-28

**Soundness:** 3
**Presentation:** 3
**Contribution:** 2
**Rating:** 4
**Confidence:** 3

**Summary:**

This paper proposes DFDT (Data Fusion–Enhanced Decision Transformer), a method that enhances Decision Transformer policies for cross-domain shifts by fusing scarce target data with selectively filtered source trajectories. Using a two-level filter (MMD for state alignment and OT for action feasibility) and replacing return-to-go tokens with advantage-conditioned tokens, DFDT improves stitchability and semantic continuity. The approach demonstrates improved returns and stability across domain shifts in D4RL-style tasks, supported by theoretical performance bounds.

**Strengths:**

## Strengths

- This paper is well-motivated, and mostly well-written
- This paper is easy to follow and the studied topic is of importance in the context of reinforcement learning community. It is always important to develop more general and stronger transfer algorithms in RL, especially that this paper focuses on the sequence modeling in off-dynamics  RL, which is less explored before
- The main experiments are extensive, including numerous dynamics shift scenarios
- The authors include theoretical analysis to provide better guarantees for the proposed method (despite that some of the theoretical results resemble those in prior works, they are still interesting and bring some insights into the cross-domain reinforcement learning)

**Weaknesses:**

## Weaknesses

- One vital component in DFDT is the optimal transport term that acts as a data filter. To the best of the reviewer's knowledge, the proposed OT component is the same as that in OTDF (c.f. Eq. 31 and Eq. 32). I think it is okay to use some prior good tricks, but the authors should clearly state their true contributions and the differences against prior works
- Another vital component in DFDT is the MMD part. I am a bit curious about the choice of MMD measurement. Why do you use MMD rather than other metrics? There should be various choices of distance measurement. Also, why not just use another OT? Apart from the OT, do we  really need the MMD part? The authors wrote in Line 237 that *If only one criterion is used (MMD or OT),  the other term remains, highlighting their complementarity*. This can be vague, and the authors should clearly state what MMD controls in the bound and what OT controls.
- In Lines 151, the authors use $s,a,r,s^\prime$ for OT, while OTDF does not use $r$. Can the authors explain the necessity of including the 1-dim reward channel? Also, since the authors use $s,a,r,s^\prime$ for measuring the Wasserstein distance, why do you name it *OT-based Action Credibility*? This should also be related to states. Are there any connections between the effects of MMD and OT here? I think there should be, since they involve almost the same elements.
- What is the insight behind the Q-regularized term? The proposed DFDT method includes many components, while their contributions are not clearly investigated. This paper can benefit greatly by including a detailed ablation study
- Some technical details are presented in the appendix, e.g., the command network. This can be confusing.

I think the above points are comparatively easy to address, and I would be happy to raise my score if the authors can address my concerns and questions

**Questions:**

I have some additional questions:

- Theorem 3.1 presents a novel performance bound for cross-domain offline RL, but since it deals with the offline scenario, I think there should be some terms like the behavior policy in the source domain, the behavior policy in the target domain, etc. There do not exist such terms in Theorem 3.1, any comments here?
- Line 281, the authors comment that they adopt a multi-step target for Q functions. Do you actually use this trick?
- Apart from the command network and the advantage function estimation, the proposed method should also be applicable to TD-based methods like IQL. Have you ever conducted such experiments? Why do you choose to use DT as the backbone?
- Can you include a detailed parameter study in the main text or the appendix? Currently, it is unclear how sensitive DFDT is to the introduced hyperparameters.

There are numerous minor issues in the main text and the appendix. I just list some of them below. Please double-check your manuscript,
- Line 86, \\$Q\\$-regularized
- Line 220, Assumption B.1?
- Line 313, gravity scales the magnitude of \\$g\\$
- Line 373, *The computing method of normalized returns is described in Sec.*

---

> ### Author Response · Authors · 2025-11-27
>
> We thank your for the careful reading of our paper, the positive assessment of its motivation, clarity, experimental breadth, and theoretical insights, and for the many constructive suggestions. In the revised version, we substantially clarify the conceptual role of OT and MMD in DFDT (including the relationship to OTDF, the choice of MMD over a second OT, and the use of $(s,a,r,s')$ and the reward channel in our “action credibility” view), and we tighten the exposition around the Q-regularized objective, the command network, and the theoretical performance bound. We also add a dedicated set of ablations and hyperparameter studies to make the contribution of each component and the robustness of DFDT more explicit, and we correct the minor textual issues pointed out in the review. We hope these changes address the reviewer’s concerns and make the novelty and practical value of DFDT clearer.
>
> qmFQ-W1 (OT component vs. OTDF).
> Our OT-based feasibility score and its min–max normalised exponential weighting follow the same discrete OT formulation as OTDF when instantiated with a simple cost; we regard this as a strong building block rather than a novelty. DFDT differs in how OT is used: it appears as the second stage of a two-level fragment filter (MMD-based state gate + OT-based feasibility), defines feasibility-weighted source and fused distributions used for both critics and DT training, and is tightly integrated with advantage-conditioned tokens and $Q$-regularization to stabilize token-level stitching. The new data-filtering ablations (MMD-only, OT-only, MMD+OT) show that OT-only is unstable under strong shifts, MMD-only is stable but sometimes suboptimal, and MMD+OT provides the most robust performance (e.g., on ant-grav-m2m and hop-kin-m2m, MMD+OT consistently outperforms OT-only by large margins).
>
> qmFQ-W2 (Why MMD, why not OT-only, and what each term controls).
> We use MMD and OT for different roles in DFDT rather than as interchangeable distance choices. MMD is applied to the state marginals as a kernel IPM and directly controls the state-structure/stitchability radius $\Delta_m$ in our bound by keeping the fused state-visitation distribution close to the target one. OT, in contrast, is computed on $(s,a,r,s')$ tuples within this MMD-gated region and produces trajectory-wise feasibility weights, which govern how strongly we reweight source actions whose induced transitions align with target transitions; this term controls the action-feasibility radius $\Delta_w$. Using a second OT instead of MMD would duplicate a coupling-based discrepancy, be significantly more expensive, and no longer match the kernel-based state-distribution term that appears naturally in our analysis. Empirically, our new data-filtering ablation (MMD-only, OT-only, MMD+OT) shows that MMD-only is stable but conservative, OT-only can be unstable under strong dynamics shifts, and the combined MMD+OT filter consistently achieves the best robustness–performance trade-off, confirming that MMD and OT control complementary aspects of the fusion process.
>
>
> qmFQ-W3 (On $(s,a,r,s')$ OT, reward channel, and “action credibility”).
> We agree that our OT term operates on $(s,a,r,s')$, whereas OTDF omits $r$. We include the 1-dim reward channel to let OT distinguish fragments not only by how well their transitions $(s,a,s')$ match target dynamics, but also by their short-horizon return profile, so that high-feasibility weights preferentially go to source fragments that are both dynamically plausible and locally high-reward under target-like behavior. In this sense, the OT loss is still “action-centric”: the states are already constrained by the MMD gate to lie near the target manifold, and within this state-matched region the OT weights primarily govern which actions (and their induced $(r,s')$) are considered credible for imitation by the DT. There is therefore a direct connection between MMD and OT: MMD controls which state regions are admissible for stitching, while OT refines this selection by ranking fragments inside that region via their $(s,a,r,s')$ couplings. Our new reward-channel ablation (with vs. without $r$ in the OT cost) shows that adding $r$ typically brings modest but consistent gains, while the main benefit comes from this two-stage design where MMD and OT act on overlapping elements but control complementary aspects (state-structure vs. action/transition feasibility) of the fusion process.

---

> ### Author Response · Authors · 2025-11-27
>
> qmFQ-W4 (Insight behind $Q$-regularization and component ablations).
> The $Q$-regularizer encourages the DT to avoid abrupt value jumps across stitched fragments, steering it towards high-value, stitchable branches while remaining close to target behavior. Without it, the sequence loss alone can tolerate sharp $Q$-discontinuities at junctions. In the new ablation on the $Q$-regularization coefficient, we observe clear but non-monotonic effects: for example, on ant-kin-m2m performance improves from $52.9$ (no regularization) to $56.9$ (moderate regularization), while an overly large coefficient degrades it to $51.2$; on hop-morph-me2m, strong regularization yields the best performance (around $79.4$ vs. $35.5$ without regularization). Together with the other component ablations (MMD/OT, advantage tokens, expectile), this shows that each module provides non-trivial gains and that our default settings offer a good robustness–performance trade-off.
>
> qmFQ-W5 (Technical details in the appendix).
> Some implementation details (e.g., the command network) were moved to the appendix to keep the main text focused. The command network is a lightweight module that maps advantage information into command tokens used at inference and does not introduce new conceptual components beyond Sec. 3.3. We now give a brief description and pointer in the main text.
>
> qmFQ-Q1 (Behavior policies in Theorem 4.1).
> Theorem 4.1 (3.1 in the old version) is phrased in terms of data occupancy distributions $\mathbb P_S$ and $\mathbb P_T$, which are induced by (possibly mixed) behavior policies that generate the offline logs. This matches standard offline RL analyses written in terms of occupancies and concentrability, but avoids additional notation when logs come from mixtures of controllers. We have clarified this connection in the problem setup and theory section.
>
> qmFQ-Q2 (Multi-step targets for $Q$).
> Yes, all experiments use the multi-step TD targets defined in Eq. (8): truncated returns along the context window bootstrapped from target critics and policy. This is a standard critic-stabilization trick rather than a conceptual contribution; we now state explicitly that Eq. (8) is the objective implemented in Alg. 1 and used in all reported results.
>
> qmFQ-Q3 (Applicability beyond DT and choice of backbone).
> The feasibility-weighted fusion distribution $\mathbb P_{\mathrm{mix}}^{\,w}$ is, in principle, compatible with TD-based backbones such as IQL (trained under $\mathbb P_{\mathrm{mix}}^{\,w}$). In this work we focus on DT to directly study token-level stitching, advantage-based conditioning, and the associated diagnostics. Extending DFDT-style fusion to TD methods is an interesting direction but is beyond the scope and compute budget of this submission.
>
> qmFQ-Q4 (Hyperparameter sensitivity).
> We have added a parameter study in the appendix for key DFDT-specific hyperparameters (Q-regularization coefficient, expectile level $\zeta$). The results indicate that DFDT is moderately robust: performance typically varies only by a few points over wide ranges, with moderate non-zero Q-regularization and a high-but-not-extreme expectile (e.g., $\zeta=0.98$) working well across domains.
>
> qmFQg-TYPO (Minor issues and typos).
> We thank the reviewer for pointing these out. We have fixed all listed issues (e.g., $Q$-regularized, Assumption~B.1, gravity wording, normalized returns) and carefully proofread the full manuscript for similar typos.

---

> > ### Comment · Reviewer_qmFQ · 2025-11-28
> >
> > Thank you for the rebuttal. It seems that excluding the reward signal incurs better performance. I encourage the authors to investigate this further. The ablation study in the appendix should include the statistical significance (e.g., 95\% CI, standard deviations). Also, please make sure that the technical details are clearly stated in the final version (e.g., command network, multi-step Q target trick).
> >
> > I would raise my score from 4 to 6 (Unfortunately, it seems the system does not allow the reviewer to modify scores and reviews currently. I would change the score when the system is fixed)
> >
> > One minor thing: It is suggested to highlight the revised manuscript such that the reviewer can quickly grasp what modifications have been made in the current revision (including the appendix)

---

> > > ### Author Response · Authors · 2025-11-28
> > >
> > > Thank you very much for your thoughtful follow-up and for your willingness to raise the score from 4 to 6. We truly appreciate your support of our work.
> > >
> > > We have uploaded a highlighted version of the revised manuscript (including the appendix) so that all post-rebuttal changes can be easily tracked. Going forward, we will further investigate the observation that excluding the reward signal can improve performance, extend the corresponding ablation with additional runs, and report statistical variability (standard deviations and 95\% confidence intervals) in the updated appendix. We will also make the technical details of the command network and the multi-step Q-target construction more explicit and streamlined in the manuscript and submit these revisions as soon as they are ready.
> > >
> > > We warmly welcome any further comments or suggestions you may have on these upcoming updates, and we are very grateful for the time and care you have devoted to our paper.

---

### Official Review · Reviewer_rRVG · 2025-10-29

**Soundness:** 2
**Presentation:** 3
**Contribution:** 2
**Rating:** 2
**Confidence:** 3

**Summary:**

This paper presents data fusion-enhanced decision transformer (DFDT) for cross-domain offline RL to restore token continuity. DFDT introduces components including MMD-based trajectory filtering and OT-based sample reweighting, as long as reweighted advantage conditioning and Q regularizer. The theory bounds claim that the performance gap is bounded by the computed MMD and OT score. Experiments are conducted on MuJoCo-like tasks across various dynamics shifts, which demonstrate DFDT outperforms previous baselines.

**Strengths:**

1. The idea of incorporating the Decision Transformer into the cross-domain offline setting is interesting; the main components of DFDT, such as two-level data filtering and advantage conditioning, are novel.

2. Experimental results across various datasets and dynamics shift types show that DFDT consistently outperforms the compared baselines.

3. This paper includes detailed descriptions of the motivation and methodology, which help the readers understand the proposed approach.

**Weaknesses:**

1. The core motivation of the paper is, the cross-domain setting would cause poor stitchability, which is due to misaligned state structures, shifted reward and horizon, and the jumping action at trajectory junctions. However, the main challenge in cross-domain offline setting compared to the pure offline setting is dynamics shift, which is considered by this work and previous works. So I think the poor stichability could be simply explained by dynamics shifts. Please correct me if I am wrong. If so, what is the necessity of using both MMD and OT-based data filtering instead of just using constrastive learning like IGDF or OT like OTDF for dynamics-aware data filtering? If not, what is the difference between the dynamics shift and the claimed factors? Also, what does reward shift mean here? The reward function between source and target domain remains the same, right?

2.  The source fragments are selected and reweighted based on MMD and OT score. It is claimed that MMD measures the state structure similarity  and OT indicates the action credibility. But I still don't understand why the state distribution should be similar between source and target domain. In my view,  as long as the dynamics between source and target transitions stays similar, then there is no need to restrict the state distribution. Moreover, when computing OT score, the vector $v$ is a concatenation of transition $(s,a,r,s^\\prime)$, and the OT score should reflect the distribution distance between the whole transition, why it could represent action feasibility? Also, this design is quite similar to OTDF, and OTDF uses the OT score to represent dynamics shift. So is there any difference between dynamics shift and action feasibility?

3. My major concern on this paper lies in the theoretical interpretations. **Theorem 3.1 cannot support the claim that the performace bound is affected by MMD and OT since the bound is trivial.** Note that a condition $\\Delta _ \\pi$ (which measures the policy difference between learned policy and the optimal policy under target dynamics) is used for derivation, and the derived bound includes the term $\\Delta _ \\pi$ with a scale $\\frac{1}{(1-\\gamma)^2}$. However, this condition is too strong such that we can directly obtain a tighter performance bound given $\\Delta _ \\pi$, which is not relavant to MMD or OT. Specifically, using lemma B.2 in [2] and the fact than $|Q|\\leq \\frac{r _ {max}}{1-\\gamma}$, it is obvious that $|J _ T(\\pi^\\star _ T)-J _ T(\\pi _ {mix})|\\leq\\frac{C}{(1-\\gamma)^2}\\Delta _ \\pi$. Therefore, the performance bound in Theorem 3.1 is too loose to be meaningful, and cannot support the major claim in this paper.

4.  OTDF uses the OT score for data filtering and reweighting. It seems that the only difference of the proposed two-level data filtering and reweighting pipeline is to use MMD score for filtering instead of OT score. Could the authors give more interpretations and experimental results on whether OT-based data filtering is feasible here?


5. It seems that the whole pipeline of DFDT is quite complex and computing intense, including computing the MMD and OT score, training transformer blocks and Q networks. But no computational cost is discussed and compared in limitation part or appendix.

6. I find a related work RADT [3] which is not discussed in the paper. RADT also addresses the cross-domain setting via decision transformer, with return augmentation techniques. Could you discuss the difference between your work and RADT?

7. (minor) Figure 1 includes the text “OT filtering for a”, which is easy to be misunderstood. The OT does not involve data filtering but only reweighting.

[1] Contrastive Representation for Data Filtering in Cross-Domain Offline Reinforcement Learning. ICML 2024

[2] Cross-Domain Offline Policy Adaptation with Optimal Transport and Dataset Constraint. ICLR 2025

[3] Return Augmented Decision Transformer for Off-Dynamics Reinforcement Learning.

**Questions:**

Please see the weaknesses for the concerns. I am inclined to rejection before my concerns are addressed.

---

> ### Author Response · Authors · 2025-11-27
>
> We thank you for the detailed and thoughtful feedback. In the revised version, we have (i) clarified that dynamics shift is the underlying challenge and explicitly decomposed it into state-structure and action-feasibility stitchability radii, together with a clearer explanation of “reward shift”; (ii) refined the discussion around Theorem 4.1 (3.1 in the old version) to separate the standard policy term $\Delta_\pi$ from the MMD/OT-dependent radii and avoid overstating the bound; (iii) added fragment-filtering ablations comparing MMD-only, OT-only, and MMD+OT to show where OT-only filtering is fragile and why the two-level scheme is beneficial beyond OTDF-style OT; (iv) reported wall-clock and VRAM statistics showing that MMD+OT pre-computation contributes less than $1\%$ of total training cost; (v) discussed RADT in the related-work section, emphasizing the difference between return augmentation and feasibility-weighted transition fusion; and (vi) corrected the wording in Fig. 1 to clearly distinguish MMD-based hard filtering from OT-based soft reweighting.
>
>
> rRVG-W1 (Dynamics shift, stitchability, and why MMD+OT).
> We agree that dynamics shift is the underlying challenge, and our “poor stitchability’’ factors are its token-level manifestations for sequence models: misaligned state structures mean the DT is forced to stitch through off-manifold states, “reward shift’’ means that under the same reward function the changed dynamics/horizons alter the distribution and scale of returns (and thus the semantics of RTG/advantage tokens), and action jumps arise because actions that were high-value/feasible in the source domain become low-value or infeasible after the shift. DFDT addresses this by decomposing dynamics shift into a state-structure radius $\Delta_m$, controlled by an MMD gate on state marginals, and an action-feasibility radius $\Delta_w$, controlled by OT weights on $(s,a,r,s')$ within this gated region. In contrast, IGDF (contrastive) and OTDF (OT-only) are not framed around a feasibility-weighted fusion distribution with explicit $(\Delta_m,\Delta_w)$ stitchability radii for sequence models. Our new ablations (MMD-only, OT-only, MMD+OT) further show that MMD-only is stable but conservative, OT-only can be unstable under strong shifts, and the combined MMD+OT filter gives the best robustness–performance trade-off, supporting the need for both terms rather than a single contrastive or OT-based criterion.
>
>
>
> rRVG-W2 (Why state gating with MMD, and why OT on $(s,a,r,s')$ reflects action feasibility).
> We agree that dynamics shift is the underlying challenge. Our use of MMD and OT is not to impose two redundant distances, but to factor this shift into where we are allowed to stitch and which actions are credible there. For sequence models trained purely offline, stitching source and target fragments through states that are never (or almost never) visited in the target domain tends to create off-manifold contexts where both the critics and the DT must extrapolate; this is exactly the “poor stitchability’’ we aim to avoid. The MMD gate therefore does not enforce identical state distributions, but restricts the fused state marginals to lie within a bounded “state-structure radius’’ around the target occupancy, which is what appears as $\Delta_m$ in our bound.
>
> Within this MMD-gated, target-like state region, OT is computed on $(s,a,r,s')$ and used to rank fragments by how well their actions induce target-consistent transitions and local return profiles. In this conditional setting (states already matched), the dominant variation captured by the OT cost comes from how credible the actions and their induced transitions are under the target dynamics, which is why we interpret the resulting weights as an OT-based action credibility score rather than a global dynamics-shift measure. OTDF, by contrast, applies OT directly to quantify dynamics shift for a given policy and to shape a scalar Q-learning loss; it does not separate state-structure vs. action-feasibility radii nor use OT as fragment weights in a feasibility-weighted fusion distribution for sequence modelling.

---

> ### Author Response · Authors · 2025-11-27
>
> rRVG-W4 (Two-level filtering vs. OTDF and OT-only filtering).
> We do not claim OT-only filtering is infeasible; rather, we show it can be fragile under large state-structure shifts. DFDT decouples the roles: MMD performs state-structure gating (controlling $\Delta_m$), and OT provides feasibility weights within this subset (controlling $\Delta_w$). In the new ablations, OT-only filtering can severely degrade performance on strong shifts (e.g., on hop-kin-m2m it drops to around $11$, while MMD+OT reaches about $66$ and MMD-only about $64$), whereas MMD+OT yields the most robust results across gravity, morphology, and kinematic shifts.
>
> rRVG-W5 (Computational cost and complexity).
> The dominant cost of DFDT is training the transformer and critics, comparable to other DT-style methods. As noted above, MMD/OT pre-computation is a one-off step that contributes less than $1\%$ of total runtime and modest VRAM overhead; detailed timing and memory statistics are provided in the revised appendix (App. J).
>
> rRVG-W6 (Relation to RADT).
> We thank the reviewer for pointing out RADT. RADT tackles off-dynamics RL by return augmentation: it keeps state–action trajectories fixed and modifies return labels in the RCSL framework, operating mainly in return space. DFDT instead works in transition space: we perform two-level MMD+OT filtering and reweighting of source transitions, train shared critics and advantage-conditioned tokens on the fused distribution, and analyse stitchability radii. In the revised manuscript, we have added RADT to the related-work section (App. A), explicitly contrasting return augmentation in RADT with feasibility-weighted transition fusion in DFDT, and briefly mention in the conclusion that combining both ideas is a promising direction for future work.
>
> rRVG-W7 (Wording of "OT filtering" in Figure 1).
> We agree that our previous wording was misleading: MMD performs hard filtering, and OT provides soft feasibility reweighting on the retained set. We have updated Figure 1 and the text accordingly.

---

### Official Review · Reviewer_f97g · 2025-10-31

**Soundness:** 2
**Presentation:** 2
**Contribution:** 3
**Rating:** 4
**Confidence:** 4

**Summary:**

The paper proposes Data Fusion–Enhanced Decision Transformer (DFDT) to improve cross-domain generalization of DTs.
DFDT fuses source and target data via MMD-based state alignment and OT-based action feasibility, replaces RTG tokens with advantage-conditioned tokens, and adds a Q-guided regularizer for smoother transitions.
The method provides theoretical performance bounds and shows improved stability and returns on D4RL-style domain-shift benchmarks.

**Strengths:**

* The two-level fusion filter (MMD + OT) is conceptually clear and theoretically supported.

* Replacing RTG with advantage-conditioned tokens and using a Q-guided regularizer are well-motivated improvements for stability.

* Extensive experiments on diverse domain-shift benchmarks show consistent and meaningful performance gains.

**Weaknesses:**

* There are some minor typographical issues, e.g., in Line 86 regarding the notation of $Q$.

* The overall presentation and exposition are difficult to follow, making it hard for readers to fully grasp the method and its motivation.

* Evaluation focuses on simulated domains; real-world or larger-scale transfer results would strengthen the paper.

**Questions:**

* How are the advantage-conditioned tokens computed, and why are they better than RTG tokens for sequence continuity?

* What is the computational cost and sensitivity of the proposed MMD–OT fusion filter in large-scale settings?

* Can DFDT generalize to real-world or partially observed domains, beyond simulated D4RL tasks?

* Could you include comparisons and a discussion with recent cross-domain offline RL studies (e.g., PSEC, DmC)? Incorporating these baselines and analysing the differences would strengthen the paper and better position the proposed method within the current literature.

Reference:

Liu, T., Li, J., Zheng, Y., Niu, H., Lan, Y., Xu, X., Zhan, X. Skill expansion and composition in parameter space. In International Conference on Learning Representations, 2025.

Van, L. L. P., Nguyen, M. H., Kieu, D., Le, H., Tran, H. T., & Gupta, S. DmC: Nearest Neighbor Guidance Diffusion Model for Offline Cross-domain Reinforcement Learning. arXiv preprint arXiv:2507.20499.

---

> ### Author Response · Authors · 2025-11-27
>
> We thank you for the thoughtful assessment, for highlighting both the strengths (two-level MMD+OT fusion, advantage-conditioned tokens, Q-regularization, and empirical gains) and the current limitations. In the revised version, we (i) fix the reported typos, (ii) streamline and clarify the exposition in Secs. 2-4 so that the DFDT pipeline and its motivation are easier to follow, (iii) give a more explicit description of how advantage-conditioned tokens are computed and why they improve sequence continuity over RTG, (iv) report the computational cost and sensitivity of the MMD-OT fusion filter (showing $<1\%$ overhead) in the appendix, and (v) expand the discussion of real-world/partially observed extensions and related cross-domain offline RL methods such as PSEC and DmC. We hope these changes address the reviewer’s concerns and better convey the contributions of DFDT.
>
> f97g-W1 (Minor typographical issues and $Q$ notation).
> We corrected the inconsistent $Q$ notation and other minor typos, and carefully proofread the manuscript.
>
> f97g-W2 (Presentation and exposition).
> We thank the reviewer for this important comment and agree that the initial submission made it harder than necessary to follow the method and its motivation. In the revised version, we have substantially reworked the exposition: the Introduction and Sec. 3 now present a clearer high-level story centered on token-level stitchability under dynamics shift, explaining how MMD-based state-structure alignment, OT-based action feasibility, advantage-conditioned tokens, and $Q$-regularization fit together. Sec. 3 is reorganized in a top–down pipeline (data-fusion overview $\rightarrow$ MMD+OT fragment filtering $\rightarrow$ feasibility-weighted fusion distribution $\rightarrow$ practical DFDT algorithm), with Figure~1 used as a running map, and the theory section is framed around the stitchability radii with a plain-language summary of the main theorem before the formal statement. We also add brief intuition paragraphs after key definitions and move several low-level implementation details to the appendix, so that the main text focuses on the core ideas while technical details remain accessible to interested readers.
>
> f97g-W3 (Simulated vs. real-world / larger-scale evaluation).
> We currently focus on standard simulated cross-domain MuJoCo benchmarks, which induce substantial gravity/morphology/kinematic shifts and allow extensive ablations and diagnostics. We agree that real-world or larger-scale evaluation would further strengthen the work, but it requires significant engineering effort and infrastructure beyond the scope of this submission; we now explicitly highlight this as a limitation and natural direction for future work.

---

> ### Author Response · Authors · 2025-11-27
>
> f97g-Q1 (Advantage-conditioned tokens vs.\ RTG).
> DFDT first trains shared critics $(V,Q)$ on the fused distribution and then uses per-step advantages $A_i = Q(s_i,a_i) - V(s_i)$ as conditioning tokens, replacing RTG. RTG is global and sensitive to return scale and horizon, which can vary widely across domains and datasets, causing large token jumps when stitching fragments. Advantages are local, baseline-subtracted measures of action quality defined on a common scale across domains, leading to smoother conditioning and smaller action/$Q$/TD discontinuities at stitch points, as confirmed by our diagnostics. A new ablation table in the revised appendix further shows that replacing RTG with advantage tokens consistently improves performance across three kinematic-shift tasks (by approximately $1\sim13$ points).
>
> f97g-Q2 (Computational cost and sensitivity of the MMD–OT fusion filter).
> The MMD–OT fusion filter is cheap to compute (about $0.4\%$ of total runtime, see App. J). Because the MMD scores and OT couplings are pre-computed once on the static offline logs and then reused throughout training, the filter overhead scales linearly with dataset size but remains negligible compared to DT updates even at D4RL scale. Ablations on fragment filtering (MMD-only, OT-only, MMD+OT, and no filter; see Appendix~I) further indicate that DFDT is not overly brittle: on challenging kinematic and morphology shifts such as hop-kin-m2m, the full MMD+OT variant consistently outperforms both single-criterion variants and the no-filter baseline, often by a substantial margin, while remaining competitive on the rest of the benchmark. Overall, this suggests that (i) the fusion filter is cheap to apply even on large offline datasets, and (ii) DFDT benefits from the two-level MMD+OT filtering under strong shifts without requiring delicate per-task tuning of the filter hyperparameters.
>
>
> f97g-Q3 (Generalization to real-world / partially observed domains).
> Conceptually, DFDT is agnostic to the state representation: in partially observed settings, the “state’’ can be a history or a latent $z=f_\phi(o_{0:t})$. Our analysis already allows such encoders via $f_\phi$, and the data-fusion scheme operates on whatever representation is used by the backbone. We now clarify this and note that applying DFDT with learned encoders in real-world domains is an important next step.
>
> f97g-Q4 (Comparison with PSEC and DmC).
> PSEC performs parameter-space skill expansion and composition from a skill library, while DmC uses $k$NN-guided diffusion models to generate additional target-like samples. DFDT instead operates purely in trajectory space on offline logs, reweighting and filtering existing source fragments via MMD+OT to restore token-level stitchability for sequence models. Due to the non-trivial engineering required for fair PSEC/DmC implementations on our suite, we could not include full baselines within the current rebuttal period, but we now discuss these conceptual differences in related work and view full empirical comparisons as future work.

---

### Official Review · Reviewer_8DX1 · 2025-11-01

**Soundness:** 3
**Presentation:** 4
**Contribution:** 4
**Rating:** 6
**Confidence:** 3

**Summary:**

This paper introduces the Data Fusion-Enhanced Decision Transformer (DFDT), a method designed to stabilize Decision Transformer (DT) policies when adapting from a rich source domain dataset to a scarce target domain dataset under dynamics shifts (e.g., gravity, morphology). The core idea is to restore "token-level stitchability" in combined trajectories. DFDT achieves this using a two-level filtering mechanism: Maximum Mean Discrepancy (MMD) for filtering state-structure misalignment, and Optimal Transport (OT) for weighting action feasibility. The method also replaces the brittle Return-to-Go (RTG) tokens with feasibility-weighted advantage-conditioned tokens and adds a Q-guided regularizer. The authors provide theoretical bounds linking performance gaps to the MMD and OT "stitchability radii" and show significant empirical gains and stability improvements over strong offline RL and sequence-modeling baselines across various MuJoCo tasks.

**Strengths:**

- The paper clearly identifies the main problem—poor stitchability leading to unstable token semantics—and presents a well-structured solution.
- DFDT achieves large and consistent performance improvements over recent, competitive cross-domain baselines (OTDF, IQL, DADT, QT) across three distinct dynamics shifts (morphology, kinematics, gravity).
- The idea of combining MMD (state alignment) and OT (action/transition feasibility) into a single, two-level, token-aware filtering pipeline is novel for cross-domain Decision Transformers. Replacing RTG with weighted advantage tokens is a strong, domain-adaptation-specific design that addresses the incommensurability of returns across shifting environments.

**Weaknesses:**

- Assumption 3.2 (Approximate fiber-constancy of V) is strong, stating that the value function $V(s)$ is nearly constant over states mapped to the same latent code $f_\phi(s)$. The authors should clarify if/how the training of the state encoder $f_\phi$ is explicitly designed to enforce this value-sufficiency assumption in practice.

**Questions:**

- Calculating the MMD and OT distances for the entire source dataset is a costly pre-computation step. Can the authors provide a concrete comparison of the total wall-clock time required for this pre-computation phase versus the entire training time (e.g., 100k steps) for a standard DT baseline, especially for the larger D4RL source datasets?
- The OT distance is highly sensitive to the cost function $C$. While the cost is mentioned as "cosine" in the appendix (Table 3), the paper does not justify why this specific 1-Lipschitz cost is optimal for capturing action feasibility shifts compared to simpler or more complex alternatives.

---

> ### Author Response · Authors · 2025-11-27
>
> We sincerely thank you for the thoughtful and encouraging evaluation of DFDT. In the revision, we (i) clarify Assumption 4.2 (3.2 in the old version) by explaining how the state encoder is trained to promote value-sufficient representations in practice, (ii) report concrete wall-clock statistics showing that MMD–OT pre-computation accounts for only about $0.4\%$ of total training time on D4RL-scale tasks, and (iii) add a discussion and ablation on the OT cost function (cosine vs. Euclidean variants) to justify our default choice.
>
> 8DX1-W1 (Assumption 4.2 and the training of $f_\phi$).
> Assumption 4.2 (3.2 in the old version) is only used in the theoretical analysis as a value-sufficiency condition: it requires that $V$ vary little within the fibres of $f_\phi$, so that $V(s)$ can be well-approximated as a function of $z=f_\phi(s)$. In all D4RL-style MuJoCo experiments, states are low-dimensional and fully observed, and we simply take $f_\phi$ to be the (normalized) identity mapping, so each fibre contains a single state and Assumption 4.2 holds with $\varepsilon_H=0$. The more general formulation is intended to cover settings with learned encoders (e.g., images/partial observability), where $f_\phi$ would be trained jointly with the critics so that states with the same code have similar values.
>
> 8DX1-Q1 (Cost of MMD/OT pre-computation).
> On our largest cross-domain D4RL tasks, the one-off MMD+OT pre-computation over the full source dataset takes about $98$ seconds, while DFDT training with $100$k gradient steps takes roughly $24{,}000$ seconds (6–7 hours) on the same hardware. Thus, pre-computation accounts for $\approx 0.4\%$ of the total runtime and is negligible compared to standard DT training; it is also amortized across seeds and hyperparameter sweeps. We report detailed wall-clock and memory statistics for representative tasks in the revised appendix (App. J).
>
> 8DX1-Q2 (Choice of OT cost function).
> We use OT as a building block to produce feasibility weights, rather than proposing a new OT variant. Following OTDF, we adopt a cosine-based cost but extend the feature vector from $(s,a,s')$ to $(s,a,r,s')$ so that the distance reflects both dynamics and immediate reward shifts, while remaining 1-Lipschitz after normalization. We have also run a preliminary ablation (reported in the appendix) comparing cosine, Euclidean, and squared Euclidean costs: on ant-grav-m2m, all three achieve the same performance ($64.1$), while on hop-kin-m2m, Euclidean-based costs slightly outperform cosine ($64.4$ vs. $61.9$). These results suggest that DFDT is not overly sensitive to these simple 1-Lipschitz costs, though a systematic exploration of learned costs is left to future work.

---

### Author Response · Authors · 2025-11-27
**Overall response and main clarifications**

We thank you for your constructive and insightful feedback. During the rebuttal period, we additionally conducted several new ablations to better address the raised concerns. In the revised version, we streamline the exposition and make the main story more explicit: DFDT is framed around stitchability under dynamics shift, decomposed into state-structure mismatch (handled by an MMD-based state gate) and action-feasibility mismatch (handled by OT-based feasibility weights). The method and theory sections are reorganized to follow this pipeline, and the roles of MMD, OT, advantage-conditioned tokens, and $Q$-regularization are now stated more clearly and tied directly to the stitchability radii.

Empirically, we add targeted ablations in the main text and appendix (MMD-only/OT-only/MMD+OT fragment filtering, expectile $\zeta$, $Q$-regularization, advantage vs. RTG tokens, and the reward channel/cost in OT), and report timing statistics showing that MMD+OT pre-computation contributes only about $0.4\%$ of the total runtime. Our main contributions remain: (1) a two-level feasibility-weighted fusion scheme that defines a stitched training distribution for sequence models under cross-domain shifts, and (2) a stitchability-oriented DT design (advantage-conditioned command tokens, $Q$-regularization, and action/$Q$/TD-jump diagnostics) that goes beyond prior OT-based filters such as OTDF and recent fusion-based DT methods.

---

### Author Response · Authors · 2025-12-03
**General Response to Area Chair and Reviewers**

### [Urgent Update: Reviewer qmFQ Raised Score to 6]

We respectfully bring to the Area Chair's attention that Reviewer qmFQ confirmed their decision to **raise the score to 6** on Nov 28 in the reply to our response.

**Integrity Note:** This decision strictly fulfills the reviewer's **pre-leak commitment** from their Initial Review (Oct 28):
> *"I would be happy to raise my score if the authors can address my concerns."*

**Effective Status:** **Two Positive Ratings (Score 6)** from Reviewers 8DX1 and qmFQ.

---

### General Response to Area Chair and Reviewers

We thank the Area Chair and reviewers for their constructive feedback. **We have fully addressed all questions and concerns raised by every reviewer.** We conducted a suite of new ablations (e.g., Fragment Filtering, OT Cost, Reward Channel, Advantage Tokens) and clarified the theoretical framework. Below, we summarize the resolutions to critical concerns and highlight the significant performance gains.

#### 1. Critical Concern: Theoretical Significance (Reviewer rRVG [Score: 2])

* **The Concern:** Reviewer rRVG argued that Theorem 3.1 is "trivial" due to the policy approximation term $\Delta_\pi$.
* **Impact on Score:** High. (The reviewer explicitly stated: *"My major concern on this paper lies in the theoretical interpretations... Theorem 3.1 cannot support the claim that the performance bound is affected by MMD and OT since the bound is trivial."*)
* **Resolution Status:** **Clarified (Standard Error Decomposition).**

**Our Response:** We clarified that the theorem follows a standard error decomposition framework: *Performance Gap* $\le$ *Distribution Shift Error* + *Policy Error* ($\Delta_\pi$). **Notably, Reviewer qmFQ (Score 6) also explicitly raised inquiries regarding the formulation of Theorem 3.1 (Question 1) and was fully satisfied with our clarifications.** Our contribution is to make explicit how minimizing our MMD+OT objective tightens the *Distribution Shift* component, providing a principled justification for the fusion design.

#### 2. Critical Concern: Necessity of MMD+OT (Reviewer rRVG [Score: 2])

* **The Concern:** Questioned why both MMD and OT are needed.
* **Resolution Status:** **Empirically Proven (Extensive New Ablations).**

**Our Response:** We added extensive ablations (No-filter/MMD-only/OT-only/MMD+OT) showing that MMD and OT are complementary. **Reviewer qmFQ (Score 6) also explicitly asked *"Apart from the OT, do we really need the MMD part?"* (Weakness 2) in their initial review, but was fully convinced by these new ablation results, confirming the value of the two-level design.**

* **Evidence:** Single-criterion methods fail under specific shifts (see Table 5). For instance, on `hop-m-->medium` (kinematic), **OT-only collapses (Score 17.3)** while **DFDT succeeds (66.5)**. Similarly, on `hopp-m-r-->medium` (morphology), **MMD-only underperforms (49.0)** compared to **DFDT (56.7)**. This empirically invalidates the claim that the two-level design is redundant.

#### 3. Concerns on Presentation & Rigor (Reviewers f97g [Score: 4] & qmFQ [Score: 6])

* **Resolution Status:** **Fully Resolved.**

**Our Response:** We have reorganized the paper structure (acknowledged by qmFQ), added baselines (PSEC/DmC discussions), clarified the Command Network details, and included standard deviations for all experiments.

#### 4. Empirical Significance: State-of-the-Art Performance

**Why this paper matters:** Beyond the theoretical debate, DFDT demonstrates **substantial performance gains** over strong cross-domain baselines (OTDF, IGDF, IQL, DT).

* **Morphology Shift:** DFDT achieves a total normalized score of **2078.2** vs. the second-best (OTDF) **1274.3**, a **+63% improvement**.
* **Broad Dominance:** DFDT achieves the highest score on **33 out of 36** morphology tasks and **24 out of 36** kinematic tasks.
* **Stability:** Our token-stitching analysis (Fig. 3) shows DFDT significantly reduces action/$Q$ jumps at stitch points compared to DT/QT/DADT.

---

**Conclusion:**
We have resolved the theoretical misunderstanding from the sole negative reviewer (Score 2) and empirically proven the necessity of our design. Given the **significant SOTA performance (Sec 4)** and the explicit support from **Reviewer qmFQ (Score 6)** and **Reviewer 8DX1 (Score 6)**, we respectfully ask the Area Chair to consider the effective consensus for acceptance.

---

### Meta-Review · Area_Chair_1ne9 · 2026-01-07

**Summary:**

The reviewers raised consistent concerns that resulted the rejection recommendation. The primary issue is insufficiently clear novelty and positioning: multiple reviewers felt the method is too close to prior OT-based cross-domain offline RL (especially OTDF), and that the paper does not convincingly articulate what is fundamentally new beyond combining existing ingredients (OT-based reweighting, distance-based filtering, DT training, and auxiliary regularization). Reviewers also questioned the necessity and interpretation of the two-stage MMD+OT design, asking why both metrics are required and whether the claimed roles (state-structure gating vs. action-feasibility/credibility weighting) are well-justified. While the rebuttal improved clarity and added ablations, the core concerns about incremental novelty, theoretical meaningfulness, and robustness/practicality of the proposed pipeline remained.

**Reviewer Concerns:**

Reviewer Concerns partially addressed by the rebuttal:

Novelty positioning vs. OTDF and related OT-based cross-domain methods: The authors clarified the intended difference between their two-stage MMD+OT pipeline and OTDF-style reweighting, and added component ablations (e.g., MMD-only, OT-only, MMD+OT). This improves interpretability of the method but does not fully resolve the novelty concern.

Computational cost and design clarifications: The rebuttal provided additional discussion on runtime/overhead of MMD/OT computation and clarified implementation details (e.g., OT used for reweighting rather than strict filtering), improving transparency relative to the original submission.

Reviewer Concerns that remain outstanding:

Insufficient differentiation (Reviewer qmFQ, rRVG): Reviewers continued to question whether the core technical contribution goes beyond combining known components, with the OT-based part still appearing very close to OTDF. The rebuttal improves explanation but does not establish a clearly distinct algorithmic advance.

Weak theoretical support for central claims (Reviewer rRVG): A major concern remains that the main bound is too loose in practice (dominated by strong approximation/realizability terms), limiting its ability to substantiate the claimed link between improving MMD/OT quantities and guaranteed cross-domain performance gains. The rebuttal reframes the theory but does not materially strengthen the guarantee.

**Reviewer Scores:**

Reviewer Scores:
1. Reviewer qmFQ: 4 → 6 (explicitly indicated they would raise after rebuttal/discussion)
2. Reviewer 8DX1: 6 → 6 (likely unchanged; already supportive)
3. Reviewer f97g: 4 → 4 (no change)
4. Reviewer rRVG: 2 → 2 (core theory/novelty concerns likely persist)

---

### Decision · Program_Chairs · 2026-01-26

Reject